# Uniform Noise Distribution and Compact Clusters: Unveiling the Success of Self-Supervised Learning in Label Noise

**Pengcheng Xu**                                        *pxu67@uwo.ca*
*Western University*

**Li Yi**                                                *lyi7@uwo.ca*
*Western University*

**Gezheng Xu**                                          *gxu86@uwo.ca*
*Western University*

**Xi Chen**                                      *xi.chen11@mcgill.ca*
*McGill University*

**Ian McLeod**                                       *aimcleod@uwo.ca*
*Western University*

**Charles Ling**                                 *charles.ling@uwo.ca*
*Western University*

**Boyu Wang**[*]                                   *bwang@csd.uwo.ca*
*Western University*

**Reviewed on OpenReview:** *https://openreview.net/forum?id=LDBjgS5Ez7*

## Abstract

Label noise is ubiquitous in real-world datasets, posing significant challenges to machine learning models. While self-supervised learning (SSL) algorithms have empirically demonstrated effectiveness in learning noisy labels, the theoretical understanding of their effectiveness remains underexplored. In this paper, we present a theoretical framework to understand how SSL methods enhance learning with noisy labels, especially for the instance-dependent label noise. We reveal that the uniform and compact cluster structures induced by contrastive SSL play a crucial role in mitigating the adverse effects of label noise. Specifically, we theoretically show that a classifier trained on SSL-learned representations significantly outperforms one trained using traditional supervised learning methods. This results from two key merits of SSL representations over label noise: 1. Uniform Noise Distribution: Label noise becomes uniformly distributed over SSL representations with respect to the true class labels, rather than the noisy ones, leading to an easier learning task. 2. Enhanced Cluster Structure: SSL enhances the formation of well-separated and compact categorical clusters, increasing inter-class distances while tightening intra-class clusters. We further theoretically justify the benefits of training a classifier on such structured representations, demonstrating that it encourages the classifier trained on noisy data to be aligned with the optimal classifier. Extensive experiments validate the robustness of SSL representations in combating label noise, confirming the practical values of our theoretical findings.

---

* Corresponding author

# 1 Introduction

Label noise is ubiquitous in the real world since acquiring accurately annotated datasets is expensive and time-consuming (Patrini et al., 2017; Xiao et al., 2015; Natarajan et al., 2013). Alternatively, a large amount of annotated images can be collected from unreliable sources such as non-expert annotators and image search engines, where label noise is inevitable (Xia et al., 2020a; Li et al., 2017). Recent self-supervised learning (SSL) with contrastive learning paradigms achieved great success in learning meaningful data representations without label information (He et al., 2020; Chen et al., 2020c; Zbontar et al., 2021; Caron et al., 2020).

In SSL, any two augmented examples from the *same image* (referred to as positive pairs) are mapped to nearby locations in the embedding space, whereas two augmented images from *different images* (referred to as negative pairs) are mapped to a distant location (Oord et al., 2018; Purushwalkam & Gupta, 2020; Chen et al., 2020b). Empirical evidence demonstrates that representations learned by SSL can be easily adapted to many downstream tasks such as image classification, objection detection, segmentation, and learning with imbalanced datasets (Grill et al., 2020; Misra & Maaten, 2020; Zhao et al., 2021; Xie et al., 2021; Liu et al., 2021; Yang & Xu, 2020).

Apart from these applications, in this paper, we show how SSL representations enhance learning with label noise. Specifically, we first construct a motivating example of instance-dependent label noise, then we prove that a classifier trained on representations learned by SSL with noisy labels is optimal over clean data distribution. Then we systematically analyze the benefits of representations learned by SSL and find two merits of SSL representations: (1) Uniform Noise Distribution: The label noise **uniformly** spreads over the learned SSL representations, (2) Enhanced Cluster Structure: The learned representations exhibit a **separated** and **compact** cluster structure with respect to true labels. It increases the distance of clusters from different classes while reducing the variance of the cluster of the same class.

For point (1), we theoretically show that the label noise is uniformly distributed across the learned representations by SSL in the motivating example, which is easier to address in practice (Cheng et al., 2020; Chen et al., 2021a; Zhang et al., 2020). We further extend the relationship between label noise and the representations learned by SSL to a more general case and provide empirical validation. For point (2), we empirically and theoretically justify that representations learned by SSL exhibit a cluster structure based on true labels.

Furthermore, we demonstrate that such a structure encourages the classifier trained on noisy data to align more closely with the optimal classifier learned from clean data. In contrast, representations learned through supervised learning (SL) do not achieve uniform noise distribution or form a good cluster structure. In particular, representations learned through supervised learning still depend on noisy labels and they exhibit clusters with respect to noisy labels instead of true labels.

From the algorithmic perspective, we show that mixup over SSL representations boosts the property of cluster structure discussed in our theory. Specifically, following the common setting of SSL evaluation (Chen et al., 2020b; He et al., 2020; Grill et al., 2020), we fix representations learned by SSL and then only train a linear classifier on the frozen representations with different types of label noise methods, which show significant improvements. The main contributions are summarized as follows:

- We provide theoretical analysis to show that representations learned by SSL exhibit a more uniform and discriminative cluster structure under label noise, leading to a better classifier than supervised learning.

- We systematically show that SSL breaks the dependency of label noise and representation, resulting in uniformly distributed label noise over classes. Further, SSL enlarges the distance of clusters from different classes while tightening the cluster of the same class.

- Empirically, we show that mixup based on conventional SSL augmentations satisfies our theoretical motivations and benefits cluster structure learning. Extensive experiments validate our analysis and the effectiveness of SSL in noisy label learning.

## 2   Related Work

**Learning with Noisy Labels.** There are different approaches to addressing the issues caused by label noise. Commonly used loss functions such as cross-entropy loss (CE) are not robust to label noise. Therefore, noise-robust loss function methods are proposed to make models fit clean examples but not mislabeled examples (Zhang & Sabuncu, 2018; Wang et al., 2019; Ma et al., 2020; Englesson & Azizpour, 2021). Alternative solutions design selection strategies to improve the confidence of clean examples and filter them from noisy data(Han et al., 2018; Yu et al., 2019; Wei et al., 2020; Song et al., 2019; Yang et al., 2023). Label correction methods correct pseudo labels of noisy samples for computing their loss functions (Zhang et al., 2021; Liu et al., 2020; Yi & Wu, 2019; Zhang et al., 2020). Noise transition matrix methods estimate the underlying label noise distribution and use it to correct the noisy labels and build a robust classifier. Other solutions include using the neural symbolic system to model and reduce the noise Smirnova et al. (2022), filtering noisy labels from sample and parameter levels (Wang et al., 2023), and enhancing the label accuracy with graph fusion (Xu et al., 2022). Cheng et al. (2021) provides a study on distilling the knowledge of SSL representations to supervised learning representations. In contrast, we focus on revealing and investigating how plain SSL representation can successfully address the label noise issue.

**Self-supervised Learning.** Representations of images learned by SSL have achieved remarkable success. SimCLR (Chen et al., 2020b) requires a large batch size to contain sufficient in-batch negative pairs and domain-specific augmentations such as Gaussian blur, color distortions, and color jittering. However, a large batch size is infeasible. MoCo (He et al., 2020) solves this issue by introducing a memory bank to store representations of data from previous iterations. The later work in this series uses the optimized large batch to remove the memory bank and introduce ViT into the framework (Chen et al., 2021b). BYOL (Grill et al., 2020) and SimSiam (Chen & He, 2021) propose new frameworks without using negative pairs so they can work with a reasonable batch size. On the other hand, SSL relies on domain-specific image augmentation. That is to assume that image augmentations such as changing the colors of images should not affect labels of images in downstream tasks (Tsai et al., 2020). DACL (Verma et al., 2021) and I-MIX (Lee et al., 2020) both leverage mixup augmentation (Zhang et al., 2017) as domain-agnostic augmentation and they find that SSL methods with both domain-agnostic and domain-specific augmentations perform better.

**Noisy labels with SSL.** Applying SSL methods to mitigate label noise has recently been studied. Li et al. (2021) propose the contrastive learning loss in the principle subspace via autoencoder and the mixup in the low dimensional space to learn a robust representation. Ghosh & Lan (2021) empirically validates that the representations pretrained by SSL methods are beneficial for noisy label learning. C2D (Zheltonozhskii et al., 2022) propose a contrast and divide method to empirically address the warm-up obstacle for memorizing the noise. Yi et al. (2022) analyze the memorization issue in cross-entropy issue and propose a contrastive regularization to mitigate it. Xue et al. (2022) shows that contrastive learning boosts robustness by analyzing the singular values of the representation matrix. However, these methods either lack systematic theoretical analysis or rely on extra assumptions of sub-class structure to show how SSL benefits noisy label learning.

Different from these discussed methods, our method provides theoretical analysis that SSL methods benefit noisy label learning by learning a more separated and compact cluster structure. Concretely, we analytically show that SSL can enlarge the cluster distance from different classes and reduce the variance of each cluster. Further, we empirically prove that the built-on mixup can achieve our theoretical motivations and improve noisy label learning.

## 3   Analysis of SSL representations with noisy labels

We first provide a motivating example to show that SSL can be significantly better than supervised learning, which enables us to explore and investigate the benefits of representations learned by SSL.

We first construct a binary classification problem with two linearly separable clusters, where the samples from clusters are artificially flipped according to a label noise function. We denote $y_i$ by the true label for $x_i$

and assume it is a balanced sample from $\{-1, +1\}$. Then the instance $x_i$ is decided in the following manner:

$$x_i = \begin{cases} e_1\zeta_i + e_2\xi_i, & \text{if } y_i = +1 \\ -e_1\zeta_i - e_2\xi_i, & \text{if } y_i = -1 \end{cases}$$

where $\zeta \sim \mathcal{U}_{[0,b]}$, $\xi \sim \mathcal{U}_{[-m,b-m]}$, $b, m \in \mathcal{R}, b > m > 0$ denote the interval without the loss of generity, and $e_1, e_2 \in \mathbb{R}^d$ are two orthogonal unit-norm vectors. We assume $\beta(x,y) = \text{sign}(yx^\top e_2)$ as the instance-dependent label noise function. For each clean example $(x_i, y_i)$, the corresponding noisy example is $(x_i, \tilde{y}_i)$, where $\tilde{y}_i = y_i\beta(x_i, y_i)$. Then we can compute that there are 43.75% mislabeled examples if the noise function $\beta(x, y)$ is adopted.

We use a simple linear classifier parameterized by $\omega$ and use the gradient descent algorithm to learn the parameter $\omega$ over the noisy data $\{x_i, \tilde{y}_i\}_{i=1}^n$, with a logistic loss function. Thus, in conventional supervised learning, we have the loss:

$$\mathcal{L}(\omega) = \frac{1}{n}\sum_{i=1}^n \log\left(1 + \exp\left(-\tilde{y}_i\omega^\top x_i\right)\right).$$

In contrast, following the common practice in SSL research Chen et al. (2020b); He et al. (2020); Grill et al. (2020), we first learn a linear representation model with parameter $W \in \mathbb{R}^{1 \times d}$ from $\{x_i\}_i^n$ in a self-supervised manner. Specifically, we adopt the linear SSL objective function studied in Liu et al. (2021); HaoChen et al. (2021), which tends to pull two positive pairs $(x + \gamma, x + \gamma')$ to nearby locations in the embedding space:

$$W_{\text{SSL}} = \arg\min_{W \in \mathbb{R}^{1 \times d}} -\hat{\mathbb{E}}[(x + \gamma)^\top W^\top W(x + \gamma')] + \frac{1}{2}\left\|W^\top W\right\|_F^2, \tag{1}$$

where $\hat{\mathbb{E}}$ is an empirical expectation over the data, with $\gamma, \gamma'$ are independent and identical $\mathcal{N}(\mathbf{0}, \mathbf{I})$ random variables. Once the optimal representation model $W_{\text{SSL}}$ is obtained, we fix $W_{\text{SSL}}$ and then learn a linear classifier parameterized by $\theta$ on the top of representations with noisy labels $\{(W_{\text{SSL}}x_i, \tilde{y}_i)\}_i$. Analogous to supervised learning, we also use the gradient descent with a logistic loss function $\mathcal{L}(\theta)$ to train the classifier.

$$\mathcal{L}(\theta) = \frac{1}{n}\sum_{i=1}^n \log\left(1 + \exp\left(-\tilde{y}_i\theta^\top W_{\text{SSL}}x_i\right)\right).$$

The following theorem states the behavior of linear classifiers on input data $\{(x_i, \tilde{y}_i)\}_i$ and representations of inputs data $\{(W_{\text{SSL}}x_i, \tilde{y}_i)\}_i$, respectively.

**Theorem 3.1.** *For a linear classifier trained with logistic loss function, let $\tilde{\omega}, \tilde{\theta}$ be normalized optimal parameters via gradient descent with logistic loss over the data $\{(x_i, \tilde{y}_i)\}_i$ and $\{(W_{\text{SSL}}x_i, \tilde{y}_i)\}_i$, respectively. Then the generalization accuracy in supervised learning is upper bounded by:*

$$\Pr_{(x,y)}[\text{sign}(\tilde{\omega}^\top x) = y] \leq \frac{9}{16} + \frac{2d}{3n}, \tag{2}$$

*While the generalization accuracy in SSL is lower bounded by:*

$$\Pr_{(x,y)}[\text{sign}(\tilde{\theta}^\top W_{\text{SSL}}x) = y] \geq 1 - 2e^{-n/128}. \tag{3}$$

**Remark**  Theorem 3.1 reveals two interesting facts in the presence of label noise. (1) The prediction accuracy under SSL is guaranteed to be a high value via a provable lower bound. The lower bound could further converge to 1 (perfect prediction without error) when sample size $n \to +\infty$. (2) In contrast, in supervised learning, simply collecting more samples does not guarantee a high accuracy, where the upper bound of the accuracy converges to 9/16 as $n \to +\infty$. Theorem 3.1 is in proved in Supplementary Section B.

## 4  Why SSL Works

To show the benefits of SSL in noisy labels, we analyze the learned representation $W_{\text{SSL}}$ from Eq. (1).

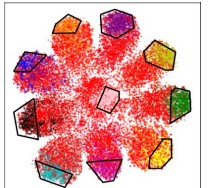 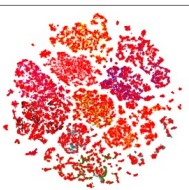 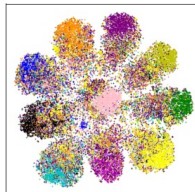 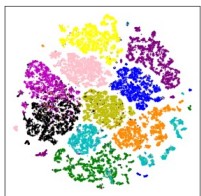 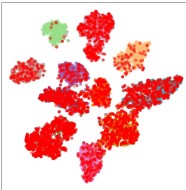 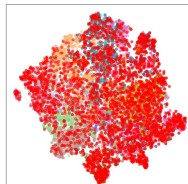

(a) SL(noisy labels)    (b) SSL(noisy labels)    (c) SL(true labels)    (d) SSL(true labels)    (e) SL(noisy labels)    (f) SSL(noisy labels)

Figure 1: T-SNE of 60% instance-dependent label noise on CIFAR-10 and Clothing1M. We train a ResNet34 on the noisy data by supervised learning (SL), and we visualize the representations learned by SL in (a) and (c) with respect to noisy labels and true labels, respectively. Similarly, we also train an SSL ResNet34 and visualize data representations in (b) and (d). We highlight regions with solid polygons that suffer from the label noise in (a), where red points (label noise) represent wrong-labeled data. In (b), the red points are almost uniformly spread over the data representations. Similarly, we also show results on the real-world Clothing1M trained with ResNet50. The similar observation shows that the label noise is not uniformly distributed in SL (e) while almost uniformly distributed in all classes in SSL (f).

**Proposition 4.1.** *Let $W_{\mathrm{SSL}}$ denote the feature representation learned from SSL and $e_1$ denote the discriminative feature-related true labels. The optimal solution $W_{\mathrm{SSL}}$ in Eq. (1) converges in probability to $ke_1$ with the constant $k > 0$.*

The solution $W_{\mathrm{SSL}}$ is the span of the vector $e_1$, which is crucial for learning an optimal classifier in Eq.(3). Note that only $e_1$ determines the true labels of data $x$. In fact, the injected label noise depends on the non-discriminative feature $e_2$ but not the discriminative feature $e_1$. If we orthogonally project data $x$ onto the direction of $e_1$, the label noise is independent of and is uniformly distributed over the projected data points spanned by $e_1$. The representation model $W_{\mathrm{SSL}}$ exactly maps the data $x$ onto the direction of $e_1$ orthogonally. Thus, the label noise is uniformly distributed over data representations $W_{\mathrm{SSL}}x$, which makes the label noise easier to address.

**Remark** The Proposition 4.1 indicates that the label noise dependent on the non-discriminative feature $e_2$ is uniformly distributed over the projected data representation $W_{SSL}$ spanned by $e_1$.

Now, we discuss the benefits of uniformly distributed noise toward learning a generalized classifier from two aspects. First, if the label noise is *uniformly distributed* across the representation, the classifier trained on data representation with this label noise can generalize better. Specifically, it can be verified that the optimal Bayes classifier, $h(x) = \mathrm{sign}(e_1^\top x)$, is also the optimal classifier over the clean distribution. On the other hand, the classifier trained with the supervised learning method from Theorem 3.1 is forced to learn spurious correlations between the inputs $e_2$ and the labels.

Second, estimating a noise transition matrix is easier when the label noise is uniformly distributed over inputs. In particular, the instance-dependent label noise can be characterized by the noise transition matrix $T(x) \in \mathbb{R}^{|\mathcal{Y}| \times |\mathcal{Y}|}$, where $T(x)_{ij}$ measures the probability of observing a corrupted label $j$ given the true label $i$ and an instance $x$. The issue of label noise is then solved by estimating the noise transition matrix $T(x)$ (Xiao et al., 2015; Liu & Tao, 2015; Patrini et al., 2017; Goldberger & Ben-Reuven, 2016). Estimating $T(x)$ for instance-dependent label noise is practically challenging since $T(x)$ can be different for different $x$ and we may need to parametrize $n$ different $T(x)$ from the noisy dataset of size $n$ by a neural network (Cheng et al., 2020; Xia et al., 2020b; Berthon et al., 2021). In contrast, $T$ is the same for all $x$ for *symmetric label noise* (i.e, label noise is uniformly distributed over data) (Patrini et al., 2017) and we only need to estimate a constant noise transition matrix. Therefore, by estimating a single noise transition matrix instead of parameterizing $n$ noise transition matrices by a neural network, the label noise is easier to solve.

## 5 Generalized Observations in Real World

In this section, we validate our theoretical analysis of the SSL representations in the real-world data. Concretely, we visualized the feature representations learned with supervised learning and SSL in Fig 1.

The data representations of supervised learning for noisy labels are in Fig 1(a) while representations of SSL are in Fig 1(b). Fig 1(a) shows that in the representations learned by supervised learning, the label noise and representations are still dependent (solid polygons highlight the regions), whereas the SSL breaks such dependency and makes the label noise uniformly distributed across the data representations. We also get similar observations on the real-world Clothing1M dataset as shown in (e) and (f). The representations of green and yellow classes have less label noise, which indicates that label noise is not uniformly distributed in the supervised learning classifier in Clothing1M. This empirical comparison and evidence in real-world data confirms our analysis about uniformly distributed noise in Proposition 4.1. Quantitatively, results in Table 6 show SSL clearly and consistently improves the performance of label noise, which concides with the implication of Theorem 3.1. These experiment results validate our experiments quantitatively and qualitatively.

Besides, we also visualize these representations with respect to their true labels in Fig 1(c-d). We find that representations learned by SSL exhibit an intrinsic cluster structure that is consistent with the true labels (Fig 1(d)). In contrast, Fig 1(c) shows that representations learned by supervised learning do not exhibit a cluster structure with respect to true labels. Thus, it further motivates us to explore how the cluster structure learned by SSL helps noisy label learning. We explore the theoretical properties of SSL on cluster structure in the next sections.

## 6 Compact Cluster Structure for Label Noise

In this section, we investigate how the cluster structure can help mitigate the label noise. Concretely, we show that for fixed representations, a good cluster structure encourages the classifier to be aligned to the optimal classifier, resulting in better generalization performance.

For simplicity, we use a two-component Gaussian mixture model to describe the clusters of representations, with each cluster representing one class. We assume that representations from class $+1$ are sampled from $\mathcal{N}(\mu, \Sigma)$ and representations from class $-1$ are sampled from $\mathcal{N}(-\mu, \Sigma)$, where $\mu \in \mathbb{R}^d$ and $\Sigma \in \mathbb{R}^{d \times d}$. In this case, the distance between two clusters is controlled by $\|\mu\|$, and the variance of each cluster is controlled by the sum of eigenvalues of $\Sigma$, which is equivalent to the trace of $\Sigma$.

We connect the cluster structure to $-\widetilde{\nabla \mathcal{L}}(\omega_0)^\top \tilde{\mu}$, which can characterize the performance of the linear classifier, where $\widetilde{\nabla \mathcal{L}}(\omega_0) = \frac{\nabla \mathcal{L}(\omega_0)}{\|\nabla \mathcal{L}(\omega_0)\|}$, $\tilde{\mu} = \frac{\mu}{\|\mu\|}$, and $\nabla \mathcal{L}(\omega_0)$ is the gradient of the logistic loss computed by the linear classifier (initialized by $\omega_0$). The normalized gradient of the loss $-\widetilde{\nabla \mathcal{L}}(\omega_0)$ represents the direction of the steepest descent in the loss function calculated on the noisy data. Given that an optimal classifier obtained from the clean data is $k\mu$ for any scalar $k > 0$, $-\widetilde{\nabla \mathcal{L}}(\omega_0)^\top \tilde{\mu}$ can measure the cosine similarity between the gradient descent direction and the direction where the optimal classifier points. After applying one-step gradient descent on the classifier, the updated classifier is more correlated to the optimal classifier if the cosine similarity is higher. More details can be found in the Appendix. This intuitively explains why $-\widetilde{\nabla \mathcal{L}}(\omega_0)^\top \tilde{\mu}$ can be used to measure the performance of the linear classifier. Now we focus on establishing the relationship between $-\widetilde{\nabla \mathcal{L}}(\omega_0)^\top \tilde{\mu}$ and the cluster structure.

As shown in Section 4, label noise is uniformly distributed over representations. Thus, we define the symmetric label noise function:

$$\beta(x, y) = \begin{cases} -1, & \text{with probability } r \\ +1, & \text{with probability } 1 - r \end{cases}$$

where $0 < r < 1$ controls the noise level. Note that for symmetric label noise, the label noise function $\beta(x, y)$ is independent of the data. The relationship between $-\widetilde{\nabla \mathcal{L}}(\omega_0)^\top \tilde{\mu}$ and the cluster structure is presented in Theorem 6.1.

**Theorem 6.1.** *Let $r$ denote the noise level of symmetric label noise function $\beta(x, y)$, if at least half of examples are clean ($r < \frac{1}{2}$),*

$$-\widetilde{\nabla \mathcal{L}}(\omega_0)^\top \tilde{\mu} \geq \sqrt{\frac{\|\mu\|^2}{c\text{Tr}(\Sigma) + \|\mu\|^2}}(1 - 2r) + o(n^{-1/3}), \tag{4}$$

*where $\text{Tr}(\Sigma)$ is the trace of $\Sigma$ and $c > 0$ is a constant.*

Theorem 6.1 provides a lower bound for $-\widetilde{\nabla \mathcal{L}}(\omega_0)^\top \tilde{\mu}$. The larger lower bound means the updated classifier is more correlated to the optimal one.

The lower bound can be affected by the noise level $r$ and the following two cluster properties: **1)** the distance between two clusters $\|\mu\|$, **2)** the variance of each cluster $\mathrm{Tr}(\Sigma)$. Without considering any label correction techniques, the noise level $r$ is fixed given a dataset. Therefore, by learning clusters of data representations that are distant from each other (larger $\|\mu\|$) and/or by learning tight representation clusters (smaller $\mathrm{Tr}(\Sigma)$), the classifier generalizes better.

**Remark** We remark that the spirits of encouraging a good cluster structure are the same for other forms of label noise such as asymmetric label noise, though their expressions of $-\widetilde{\nabla \mathcal{L}}(\omega_0)^\top \tilde{\mu}$ are different. Details are in Appendix C.2.

We empirically justify that linear classifiers get better performance when $-\widetilde{\nabla \mathcal{L}}(\omega_0)^\top \tilde{\mu}$ becomes larger. Fig 2 shows the performances of classifiers trained on data points with different cluster structures given a fixed noise level 40%. Specifically, with the same variance, the classifier trained on clusters with larger distances performs better (orange line v.s. red line). While with the same distance, the classifier trained on tight clusters performs better (orange line v.s. blue line). The two histograms show that the linear classifier with larger $-\widetilde{\nabla \mathcal{L}}(\omega_0)^\top \tilde{\mu}$ (orange) fits clean examples better, compared with the linear classifier (red). It also highlights that representations with better cluster structure help the classifier generalize better on clean data distribution.

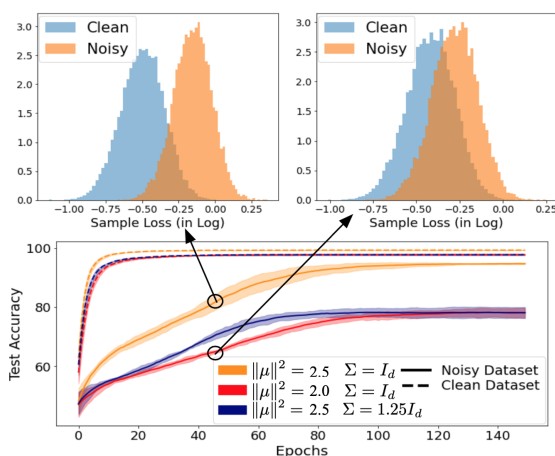

Figure 2: Linear classifiers trained on synthetic datasets with 40% noise level. The dashed line is the performance of classifiers without label noise and the solid line is that with label noise. The histograms are sample loss values at epoch = 50 with respect to whether they are mislabeled.

## 7 Benefits of SSL on Cluster Structure

In this section, we rigorously justify that the cluster properties characterized in Theorem 6.1 can be achieved by SSL. In particular, we focus on the SSL objective function Eq. (5) that has been studied in Wang & Isola (2020). Notably, the loss and its variants have been widely adopted in the SSL (Chen et al., 2020b; He et al., 2020; Tsai et al., 2021).

$$\mathcal{L}_{\mathrm{ctr}}(f) = \frac{1}{n} \sum_{i=1}^{n} \underbrace{\mathbb{E}_{\substack{u_i \sim \mathrm{Pr}(u|x_i) \\ u_i^+ \sim \mathrm{Pr}(u|x_i)}} \left[ \left\| f(u_i) - f(u_i^+) \right\|^2 \right]}_{\mathcal{L}_{\mathrm{Align}}(i)} \tag{5}$$

$$+ \lambda \log \left[ \frac{1}{n(n-1)} \sum_{i \neq j} \underbrace{\mathbb{E}_{\substack{u_i \sim \mathrm{Pr}(u|x_i) \\ u_j \sim \mathrm{Pr}(u|x_j)}} \left[ e^{-\left\| f(u_i) - f(u_j) \right\|^2} \right]}_{\mathcal{L}_{\mathrm{Uniform}}(i,j)} \right],$$

where $\lambda$ is a hyper-parameter. $\{x_1, x_2, \cdots, x_n\}$ are input instances, $P(u|x)$ denotes the conditional distribution of an augmented instance $u$ gievn $x$, and $f(\cdot)$ is a representation network that takes $u$ as an input. Intuitively, Eq. (5) minimizes the distance between two representations of different views from the same instance, and the $\mathcal{L}_{\mathrm{Uniform}}$ makes representations uniformly distributed over the embedding space.

To characterize the cluster properties studied in Theorem 6.1, we introduce the notion of $\delta$-cluster closeness.

**Definition 7.1** ($\delta$-cluster closeness). *Let $S_i$ be the support where $\mathrm{Pr}(u|x_i) > 0$ for any $u \in S_i$. $S_i$ and $S_j$ are $\delta$-cluster close if $\mathrm{Pr}[u \in S_i \cap S_j | x_i] \geq \delta$ for any two different instances $x_i, x_j$ with $y_i = y_j$.*

The notion of $\delta$-cluster closeness is similar to the cluster assumption in Lafferty & Wasserman (2007); Rigollet (2007); Singh et al. (2008). Definition 7.1 reveals that the instance augmentations should be rich enough so

that any two different distorted augmentations from the same class can be overlapped. Following (Verma et al., 2021; Lee et al., 2020), we apply mixup data augmentation on top of the conventional SSL data augmentations (Chen et al., 2020b) in order to have richer instance augmentations, making Definition 7.1 hold with a large $\delta$. The definition of $\delta$-cluster closeness provides us a tool to investigate how Eq. (5) influence the cluster properties: the distances between any two clusters and the variance of each cluster.

To analyze the cluster properties, we decompose $\mathcal{L}_{\mathrm{ctr}}(f)$ into three components and study their effects individually.

$$
\mathcal{L}_{\mathrm{ctr}}(f) = \frac{1}{n} \sum_{m \in \mathcal{Y}} \sum_{i \in J_m} \mathcal{L}_{\mathrm{Align}}(i) + \lambda \log \left[ \frac{1}{n(n-1)} \sum_{\substack{m \in \mathcal{Y}, n \in \mathcal{Y} \\ m \neq n}} \sum_{\substack{i \in J_m \\ j \in J_n}} \mathcal{L}_{\mathrm{Uniform}}(i,j) \right.
$$
$$
\left. + \frac{1}{n(n-1)} \sum_{m \in \mathcal{Y}} \sum_{\substack{i,j \in J_m \\ i \neq j}} \mathcal{L}_{\mathrm{Uniform}}(i,j) \right]
\tag{6}
$$

The following lemma indicates that a large distance between any two clusters can be achieved by optimizing $\mathcal{L}_{\mathrm{Uniform}}(i,j)$ for some $i,j$. We denote $J_y$ by the index set corresponding to true class $y$.

**Lemma 7.2.** *Let $\hat{\mu}_i = \sum_{k \in J_i} \frac{f(u_k)}{|J_i|}$, $\hat{\mu}_j = \sum_{k \in J_j} \frac{f(u_k)}{|J_j|}$ be sample means of cluster $i$ and cluster $j$ with $i \neq j$. Without loss of generality, we assume $|J_i| = |J_j|$. Then*

$$
\mathbb{E}[\|\hat{\mu}_i - \hat{\mu}_j\|] \geq -\frac{1}{|J_i|} \sum_{k \in J_i} \log \left( \mathcal{L}_{Uniform}(k, g(k)) \right)
$$

*where the function $g : J_i \to J_j$ is any bijective function, and the expectation is over the data augmentation.*

**Remark**  Lemma 7.2 indicates that the distance between the cluster $i$ and the cluster $j$ can be lower bounded by $-\log \left( \mathcal{L}_{\mathrm{Uniform}}(k, g(k)) \right)$ for $k \in J_i$. Since $-\mathcal{L}_{\mathrm{Uniform}}(k, g(k))$ measures the distance between $f(u_k)$ from cluster $i$ and $f(u_{g(k)})$ from cluster $j$, minimizing $\mathcal{L}_{\mathrm{Uniform}}(k, g(k))$ for all $k \in J_i$ increases the distance between the cluster $i$ and the cluster $j$.

On the other hand, the objective function Eq. (5) also controls the variance of each cluster. The following lemma helps us understand how the SSL objective function Eq. (5) controls the variance of each cluster.

**Lemma 7.3.** *Let $\hat{\Sigma}_y = \frac{1}{|J_y|} \sum_{i \in J_y} (f(u_i) - \hat{\mu}_i)(f(u_i) - \hat{\mu}_i)^\top$ be the sample covariance matrix. Suppose Definition 7.1 holds. Then for any fixed $\delta \in (0,1)$, we have*

$$
\mathrm{Tr}(\mathbb{E}[\hat{\Sigma}_{\mathrm{y}}]) \leq \frac{2}{\delta |J_y|} \sum_{i \in J_y} \mathcal{L}_{Align}(i),
$$

*where the expectation is over the data augmentation.*

**Remark**  Lemma 7.3 shows that the variance of cluster $y$ is upper bounded by the term $\sum_{i \in J_y} \mathcal{L}_{\mathrm{Align}}(i)$, where the variance is measured by the sum of eigenvalues for the sample covariance matrix computed by representations from the cluster $y$. In other words, a small variance of cluster $y$ can be achieved by minimizing $\sum_{i \in J_y} \mathcal{L}_{\mathrm{Align}}(i)$.

Lemma 7.2 and Lemma 7.3 have shown the effects of the first two components in Eq. (6). The last component in Eq. (6) serves as a contradiction against the first component. We note that minimizing $\mathcal{L}_{\mathrm{Uniform}}(i,j)$ for $i,j$ from the same cluster undesirably increases the variance of that cluster. This intuition is justified by the following proposition.

**Proposition 7.4.** *Suppose Definition 7.1 holds with a fixed $\delta$. Then*

$$
\log[\frac{1}{n(n-1)} \sum_{m \in \mathcal{Y}} \sum_{i,j \in J_m i \neq j} \mathcal{L}_{Uniform}(i,j)] \geq -\alpha \sum_{m \in \mathcal{Y}} \sum_{i \in J_m} \mathcal{L}_{Align}(i),
$$

*where $\alpha = \frac{2(n/|\mathcal{Y}|-1)}{\delta n(n-1)} > 0$.*

Table 1: Test accuracy on CIFAR-10 and CIFAR-100 with SYM label noise over different noise levels.

| Dataset | CIFAR-10 | | | | | CIFAR-100 | | | | |
|---|---|---|---|---|---|---|---|---|---|---|
| Noise | 20% | 40% | 60% | 80% | 90% | 20% | 40% | 60% | 80% | 90% |
| SSL+CE | 92.66±0.05 | 92.60±0.06 | 92.53±0.11 | 92.22±0.04 | 91.61±0.04 | 64.16±0.13 | 63.76±0.72 | 62.25±0.42 | 60.56±0.27 | 57.07±0.96 |
| GCE | 93.16±0.18 | 90.11±0.27 | 82.35±0.29 | 74.95±0.51 | 54.34±0.81 | 71.71±0.09 | 67.72±0.19 | 59.50±0.43 | 35.80±0.62 | 14.04±0.97 |
| +MoCo | **95.74**±0.07 | **95.67**±0.06 | **95.58**±0.04 | **95.36**±0.08 | **94.68**±0.24 | **75.21**±0.03 | **74.89**±0.08 | **73.36**±0.09 | **71.91**±0.31 | **68.22**±0.72 |
| +BYOL | **95.55**±0.02 | **95.46**±0.05 | **95.32**±0.06 | **95.11**±0.08 | **94.66**±0.16 | **73.53**±0.03 | **72.04**±0.04 | **71.43**±0.09 | **69.40**±0.20 | **65.94**±0.26 |
| CT | 93.66±0.17 | 92.22±0.16 | 70.51±0.22 | 39.75±0.88 | 27.34±0.98 | 72.69±0.14 | 68.81±0.19 | 61.15±0.28 | 16.40±0.44 | 8.22±1.46 |
| +MoCo | **95.43**±0.07 | **95.37**±0.08 | **95.19**±0.23 | **91.97**±0.80 | **87.65**±1.65 | **73.86**±0.07 | **73.37**±0.12 | **72.59**±0.41 | **67.79**±0.92 | **62.69**±2.18 |
| +BYOL | **95.13**±0.02 | **94.93**±0.04 | **94.71**±0.03 | **93.58**±0.55 | **87.35**±1.37 | **72.19**±0.05 | **71.33**±0.18 | **69.49**±0.08 | **55.55**±3.28 | **52.65**±1.22 |
| ELR | 93.53±0.10 | 93.11±0.14 | 92.22±0.16 | 85.74±0.52 | 54.27±1.06 | 69.64±0.39 | 65.16±0.30 | 60.88±0.32 | 24.92±0.52 | 10.22±0.76 |
| +MoCo | **95.88**±0.05 | **95.81**±0.04 | **95.74**±0.03 | **95.65**±0.02 | **95.60**±0.09 | **72.89**±0.39 | **72.74**±0.06 | **71.74**±0.11 | **70.47**±0.19 | **66.75**±0.22 |
| +BYOL | **95.55**±0.02 | **95.43**±0.03 | **95.30**±0.06 | **95.11**±0.05 | **95.12**±0.10 | **72.48**±0.03 | **71.73**±0.06 | **70.35**±0.10 | **68.45**±0.10 | **63.70**±0.14 |
| TCL | 93.53±0.10 | 94.73±0.11 | 93.22±0.13 | 90.34±0.29 | 87.51±0.36 | 76.24±0.26 | 73.16±0.28 | 69.45±0.33 | 62.71±0.41 | 53.58±0.56 |
| +MoCo | **95.91**±0.03 | **95.71**±0.03 | **95.82**±0.03 | **95.68**±0.02 | **95.30**±0.07 | **78.81**±0.31 | **76.79**±0.16 | **72.52**±0.11 | **71.87**±0.23 | **68.63**±0.22 |
| +BYOL | **95.59**±0.11 | **95.60**±0.13 | **95.15**±0.08 | **95.03**±0.08 | **94.95**±0.15 | **78.68**±0.08 | **76.41**±0.18 | **72.25**±0.13 | **71.65**±0.22 | **66.13**±0.24 |

Table 2: Test accuracy on CIFAR-10 and CIFAR-100 with ASYM label noise over different noise levels.

| Dataset | CIFAR-10 | | | | | CIFAR-100 | | | | |
|---|---|---|---|---|---|---|---|---|---|---|
| Noise | 10% | 20% | 30% | 40% | 45% | 10% | 20% | 30% | 40% | 45% |
| SSL+CE | 92.25±0.06 | 87.63±0.21 | 86.29±0.14 | 83.86±0.22 | 79.13±0.55 | 64.01±0.20 | 63.42±0.11 | 62.24±0.62 | 59.06±0.75 | 53.16±0.54 |
| GCE | 93.02±0.10 | 91.92±0.23 | 90.85±0.28 | 89.44±0.44 | 85.51±0.59 | 73.52±0.08 | 70.05±0.31 | 65.80±0.35 | 53.49±0.53 | 44.08±1.22 |
| +MoCo | **95.63**±0.04 | **95.37**±0.08 | **95.00**±0.31 | **93.30**±0.26 | 88.31±0.41 | **74.54**±0.04 | **73.60**±0.12 | **72.63**±0.13 | **66.27**±0.24 | **56.12**±0.68 |
| +BYOL | **95.50**±0.02 | **95.23**±0.14 | **94.83**±0.19 | **93.56**±0.36 | **90.69**±0.16 | 73.17±0.04 | **72.12**±0.09 | **70.75**±0.14 | **65.09**±0.13 | 53.96±0.50 |
| CT | 94.40±0.03 | 93.32±0.11 | 90.27±0.15 | 69.47±0.21 | 66.08±0.32 | 73.88±0.04 | 69.88±0.21 | 64.64±0.68 | 55.22±0.71 | 48.22±1.00 |
| +MoCo | **95.37**±0.07 | **95.25**±0.09 | **94.33**±0.16 | **92.29**±0.32 | **86.79**±0.52 | 73.48±0.11 | **72.02**±0.26 | **69.36**±0.41 | **63.30**±0.73 | **55.70**±1.58 |
| +BYOL | **95.48**±0.03 | **94.14**±0.72 | **94.04**±0.24 | **90.72**±0.72 | **87.33**±1.23 | 72.01±0.08 | **70.46**±0.04 | **66.22**±0.37 | 54.97±0.94 | 46.62±1.24 |
| ELR | 93.90±0.08 | 93.26±0.10 | 92.52±0.13 | 90.93±0.16 | 88.49±0.24 | 73.89±0.07 | 73.44±0.20 | 72.90±0.19 | 70.62±0.34 | 65.62±1.31 |
| +MoCo | **95.73**±0.02 | **95.69**±0.04 | **94.83**±0.12 | **92.62**±1.15 | 78.92±0.95 | **74.87**±0.05 | **74.51**±0.10 | **73.75**±0.08 | **72.26**±0.05 | **67.11**±0.28 |
| +BYOL | **95.59**±0.03 | **95.49**±0.07 | **95.40**±0.03 | **94.72**±0.04 | 86.91±1.73 | 73.44±0.05 | 72.95±0.04 | 71.81±0.05 | 69.18±0.08 | 63.27±0.12 |
| TCL | 94.10±0.20 | 93.53±0.17 | 92.52±0.11 | 91.14±0.23 | 90.91±0.36 | 78.24±0.24 | 74.77±0.28 | 73.98±0.22 | 71.62±0.52 | 67.38±0.56 |
| +MoCo | **95.18**±0.08 | **95.33**±0.09 | **95.07**±0.11 | **93.85**±0.12 | **93.60**±0.09 | **80.89**±0.29 | **78.54**±0.16 | **77.74**±0.11 | **73.88**±0.21 | **69.35**±0.25 |
| +BYOL | **95.25**±0.12 | **95.38**±0.13 | **95.10**±0.11 | **93.21**±0.09 | **93.17**±0.10 | **80.28**±0.15 | **78.12**±0.08 | **77.25**±0.11 | **73.65**±0.12 | **68.65**±0.11 |

**Remark** Proposition 7.4 indicates that minimizing the third term in Eq. (6) forces the first term to be larger, which makes the variance of clusters to be larger, where the strength is controlled by a factor $\alpha$. We note that the third term is due to $\mathcal{L}_{\text{Uniform}}$ of the instances from the same class and it cannot be eliminated since the label information is not leveraged. The constraint strength is mitigated when $\alpha$ decreases. It is small when instance augmentations are rich enough ($\delta$ is large), which also highlights the importance of data augmentations in learning SSL representations. In conclusion, our analysis for the SSL objective in Eq. (6) reveals that SSL helps enlarge the inter-cluster distance (Lemma 7.2) and reduce intra-cluster variance (Lemma 7.3) and the data augmentation can help get a better trade-off (Proposition 7.4).

# 8 Experiment

To validate our analysis of the cluster properties of SSL representations, we combine different SSL methods (i.e., MoCo and BYOL) as complementary with label noise methods. Compared to MoCov2, BYOL does not explicitly compute $\mathcal{L}_{\text{Uniform}}$ but implicitly computes it by the momentum update of the network.

**Datasets.** Following previous state-of-the-arts (Huang et al., 2023; Yang et al., 2023), we evaluate our method on CIFAR-10 and CIFAR-100 (Krizhevsky et al., 2009) with different types of label noise: symmetric (SYM), asymmetric (ASYM), and instance-dependent label noise (IDN). We also evaluate our method on two real-world datasets ANIMAL-10N (Song et al., 2019) and Clothing1M (Xiao et al., 2015).

*CIFAR-10* and *CIFAR-100* both consist of 50,000 training and 10,000 test color images whose size is $32 \times 32$. The difference is that CIFAR-10 has 10 classes containing 5,000 training and 1,000 test images for each class while CIFAR-100 has 100 classes containing 500 training and 100 test images for each class.

*ANIMAL-10N* (Song et al., 2019) is a dataset with animals of confusing appearances crawled from several online search engines including Bing and Google using the predefined labels as the search keyword. It includes 50,000 training and 5,000 test images. Since images are collected and labeled by online search engines, the resulting classification has an estimated error from 6% to 10%.

*Clothing1M* (Xiao et al., 2015) has 1 million training images and 10,000 test images from 14 classes. The noisy images are collected from online shopping websites, and there are many mislabeled samples since the labels are created by the surrounding text provided by the sellers. These texts can be coarse and induce realistic noisy labels.

**Implementations.** The implementation of MoCov2 and BYOL includes the backbone and a 2-layer MLP ad the projected head. For CIFAR datasets, the backbone is ResNet34 with a 2-layer MLP as the projection head. The input and output dimensions are 512. For ANIMAL-10N and Clothing1M, the backbone is ResNet50, and the input and output dimensions are 2048.

For the data augmentations used in MoCov2 and BYOL, we use both strong image augmentation from Chen et al. (2020b) and MixUp from Lee et al. (2020). MixUp (Zhang et al., 2017) has a hyper-parameter $\lambda$ that controls the strength of interpolation between data points, where we set $\lambda = 1$ for CIFAR datasets and $\lambda = 2$ for ANIMAL-10 and Clothing-1M. Once we train the representation network, we train a linear classifier by different label noise methods on this representation network.

The linear classifier is trained for 100 epochs using SGD, where the learning rate starts from {1, 5, 10, 20, 30} and it is reduced by a factor of 5 after 20, 30 and 40 epochs. For GCE method (Zhang & Sabuncu, 2018), its parameter $q$ is selected from {0.2, 0.4, 0.6, 0.8, 0.9}; for Co-teaching method (Han et al., 2018), the warmup parameters is selected from {5, 8, 10}; for ELR method (Liu et al., 2020), the parameter $\beta$ is selected from {0.7, 0.9} and the parameter $\lambda$ is selected from {3, 5, 7}; for TCL method (Huang et al., 2023), two augmentations are used, and the temperature $\tau$ of contrastive loss and the $\alpha$ of mixup are 0.25 and 0.1. All experiments including the SSL training are conducted on two Nivida A100.

**Baselines.** To show our analysis generally facilitating learning noisy labels, we combine frozen SSL representations with different types of algorithms: robust loss function GCE (Zhang & Sabuncu, 2018), sample selection Co-teaching (Han et al., 2018), label correction ELR (Liu et al., 2020), and TCL (Huang et al., 2023) that models and filter samples simultaneously, and the standard cross-entropy (SSL+CE). GCE designs a loss function to address memorization issues for incorrect labels. Co-teaching selects clean examples to update the neural network. ELR introduces a regularization for pseudo labels. TCL uses the GMM model to model the noisy label distribution and filter wrong labels as out-of-distribution examples.

**Main Results.** Tables 1-3 show the results for SYM, ASYM, and IDN label noise on CIFAR-10 and CIFAR-100, respectively. Table 6 shows the results on ANIMAL-10N and Clothing1M. Both MoCov2 and BYOL SSL representations can improve efficacy to a wide range of label noise methods: robust loss function methods, sample selection methods, and label correction methods. Results show that training a linear classifier on frozen SSL representations over noisy datasets is significantly better than training a whole neural network over noisy datasets. Asymmetric label noise is to flip labels between semantically-similar classes. For example, cats are inherently more difficult to differentiate from dogs than trucks. To this end, we combine IDN with ASYM to generate more realistic label noise. Specifically, we choose similar images for each class and then we flip their labels to the next class. Results are reported in Table 4.

Following Zhang et al. (2020); Chen et al. (2020a); Lee et al. (2019), semantic label noise is a type of instance-dependent label noise that follows the intuition that hard instances are more likely to be mislabeled, where the hard instances are near the decision boundary of the model. To generate the semantic label noise, we train a VGG-13 (Simonyan & Zisserman, 2014) on training datasets for 30 epochs. Following Chen et al. (2020a), we select instances with the highest mislabeling scores to corrupt. For the first case, we corrupt these instances with random labels. For the second case, we corrupt these instances with predictions of the model VGG-13. We term the former TYPE-1 label noise and the latter TYPE-2 label noise. The results for the two types of label noise are reported in Table 5. Therefore, extensive experiments have demonstrated the effectiveness of applying frozen SSL representations.

Table 3: Test accuracy on CIFAR-10 and CIFAR-100 datasets with IDN label noise over different noise levels.

| Dataset | CIFAR-10 | | | | | CIFAR-100 | | | | |
|---|---|---|---|---|---|---|---|---|---|---|
| Noise | 20% | 40% | 60% | 80% | 90% | 20% | 40% | 60% | 80% | 90% |
| SSL+CE | $91.86_{\pm0.23}$ | $90.79_{\pm0.13}$ | $89.65_{\pm0.23}$ | $87.80_{\pm0.19}$ | $80.01_{\pm1.24}$ | $65.78_{\pm0.11}$ | $63.19_{\pm0.14}$ | $61.47_{\pm0.25}$ | $58.84_{\pm0.49}$ | $54.86_{\pm0.57}$ |
| GCE | $90.05_{\pm0.29}$ | $80.35_{\pm0.34}$ | $66.94_{\pm0.51}$ | $49.38_{\pm0.66}$ | $34.49_{\pm0.97}$ | $69.58_{\pm0.16}$ | $60.48_{\pm0.32}$ | $44.63_{\pm0.62}$ | $28.34_{\pm0.84}$ | $14.18_{\pm1.29}$ |
| +MoCo | $\mathbf{95.41}_{\pm0.07}$ | $\mathbf{95.05}_{\pm0.10}$ | $\mathbf{94.35}_{\pm0.21}$ | $91.87_{\pm0.33}$ | $87.72_{\pm0.28}$ | $\mathbf{74.19}_{\pm0.04}$ | $\mathbf{72.48}_{\pm0.09}$ | $\mathbf{70.47}_{\pm0.14}$ | $\mathbf{66.67}_{\pm0.46}$ | $\mathbf{61.67}_{\pm0.48}$ |
| +BYOL | $95.22_{\pm0.08}$ | $94.80_{\pm0.09}$ | $94.26_{\pm0.25}$ | $\mathbf{92.88}_{\pm0.22}$ | $\mathbf{90.33}_{\pm0.77}$ | $72.69_{\pm0.06}$ | $70.86_{\pm0.10}$ | $68.24_{\pm0.10}$ | $65.20_{\pm0.20}$ | $60.06_{\pm0.41}$ |
| CT | $91.50_{\pm0.35}$ | $85.95_{\pm0.38}$ | $74.09_{\pm0.52}$ | $30.79_{\pm0.82}$ | $22.35_{\pm0.92}$ | $69.81_{\pm0.18}$ | $62.59_{\pm0.31}$ | $52.11_{\pm0.65}$ | $16.10_{\pm0.70}$ | $7.91_{\pm0.57}$ |
| +MoCo | $\mathbf{95.23}_{\pm0.37}$ | $\mathbf{94.68}_{\pm0.31}$ | $94.07_{\pm0.58}$ | $83.91_{\pm0.62}$ | $77.87_{\pm2.69}$ | $\mathbf{73.39}_{\pm0.22}$ | $\mathbf{72.15}_{\pm0.27}$ | $\mathbf{70.14}_{\pm0.54}$ | $\mathbf{66.26}_{\pm1.08}$ | $\mathbf{58.34}_{\pm0.68}$ |
| +BYOL | $94.97_{\pm0.07}$ | $94.52_{\pm0.18}$ | $\mathbf{94.11}_{\pm0.14}$ | $\mathbf{89.09}_{\pm0.08}$ | $71.72_{\pm1.47}$ | $71.59_{\pm0.04}$ | $70.15_{\pm0.15}$ | $66.73_{\pm0.72}$ | $57.79_{\pm0.45}$ | $49.88_{\pm1.23}$ |
| ELR | $93.54_{\pm0.04}$ | $93.20_{\pm0.18}$ | $92.07_{\pm0.20}$ | $73.27_{\pm0.55}$ | $41.39_{\pm0.80}$ | $70.11_{\pm0.32}$ | $67.16_{\pm0.70}$ | $58.11_{\pm0.67}$ | $21.96_{\pm0.74}$ | $10.28_{\pm1.07}$ |
| +MoCo | $\mathbf{95.77}_{\pm0.07}$ | $\mathbf{95.70}_{\pm0.10}$ | $\mathbf{95.65}_{\pm0.07}$ | $\mathbf{95.58}_{\pm0.04}$ | $91.35_{\pm1.91}$ | $\mathbf{72.74}_{\pm0.04}$ | $\mathbf{71.56}_{\pm0.12}$ | $\mathbf{69.69}_{\pm0.22}$ | $\mathbf{65.94}_{\pm0.59}$ | $\mathbf{59.80}_{\pm0.84}$ |
| +BYOL | $95.45_{\pm0.02}$ | $95.25_{\pm0.03}$ | $95.08_{\pm0.04}$ | $95.07_{\pm0.06}$ | $\mathbf{94.91}_{\pm0.10}$ | $72.11_{\pm0.18}$ | $70.64_{\pm0.32}$ | $68.72_{\pm0.24}$ | $63.75_{\pm0.50}$ | $57.54_{\pm0.48}$ |

Table 4: Test accuracy on CIFAR-10 and CIFAR-100 with IDN-ASYM label noise over different noise levels.

| Dataset | CIFAR-10 | | | | | CIFAR-100 | | | | |
|---|---|---|---|---|---|---|---|---|---|---|
| Noise Level | 10% | 20% | 30% | 40% | 45% | 10% | 20 | 30% | 40% | 45% |
| GCE | $87.02_{\pm0.16}$ | $77.40_{\pm0.29}$ | $68.63_{\pm0.68}$ | $57.85_{\pm0.81}$ | $54.01_{\pm0.92}$ | $71.01_{\pm0.12}$ | $62.42_{\pm0.18}$ | $52.48_{\pm0.33}$ | $44.69_{\pm0.78}$ | $40.02_{\pm0.65}$ |
| +MoCo | $\mathbf{95.20}_{\pm0.02}$ | $\mathbf{94.89}_{\pm0.05}$ | $\mathbf{93.28}_{\pm0.21}$ | $84.26_{\pm0.43}$ | $\mathbf{71.66}_{\pm1.10}$ | $\mathbf{73.90}_{\pm0.04}$ | $\mathbf{71.52}_{\pm0.57}$ | $\mathbf{69.12}_{\pm0.21}$ | $\mathbf{61.69}_{\pm0.87}$ | $\mathbf{52.17}_{\pm2.23}$ |
| +BYOL | $95.19_{\pm0.05}$ | $94.85_{\pm0.10}$ | $93.66_{\pm0.21}$ | $\mathbf{85.57}_{\pm0.37}$ | $70.44_{\pm1.44}$ | $71.58_{\pm0.14}$ | $69.61_{\pm0.13}$ | $67.22_{\pm0.08}$ | $59.33_{\pm0.77}$ | $50.37_{\pm1.57}$ |
| CT | $87.75_{\pm0.28}$ | $78.37_{\pm0.30}$ | $69.31_{\pm0.55}$ | $60.48_{\pm0.57}$ | $54.62_{\pm0.80}$ | $71.97_{\pm0.09}$ | $64.33_{\pm0.13}$ | $55.61_{\pm0.26}$ | $47.12_{\pm0.32}$ | $42.23_{\pm0.46}$ |
| +MoCo | $94.73_{\pm0.45}$ | $93.25_{\pm0.42}$ | $88.92_{\pm0.64}$ | $74.67_{\pm0.91}$ | $61.13_{\pm1.37}$ | $\mathbf{72.77}_{\pm0.32}$ | $69.61_{\pm0.49}$ | $\mathbf{66.52}_{\pm0.80}$ | $\mathbf{59.37}_{\pm1.05}$ | $50.43_{\pm1.43}$ |
| +BYOL | $\mathbf{94.88}_{\pm0.06}$ | $\mathbf{93.55}_{\pm0.07}$ | $\mathbf{90.99}_{\pm0.49}$ | $\mathbf{74.80}_{\pm0.39}$ | $\mathbf{63.67}_{\pm1.07}$ | $69.54_{\pm0.03}$ | $\mathbf{67.69}_{\pm0.12}$ | $64.44_{\pm0.19}$ | $59.23_{\pm0.32}$ | $\mathbf{51.65}_{\pm0.94}$ |
| ELR | $94.08_{\pm0.05}$ | $93.97_{\pm0.08}$ | $93.91_{\pm0.14}$ | $93.79_{\pm0.20}$ | $77.76_{\pm2.24}$ | $72.21_{\pm0.23}$ | $71.96_{\pm0.24}$ | $71.83_{\pm0.41}$ | $\mathbf{70.96}_{\pm0.43}$ | $\mathbf{67.33}_{\pm0.45}$ |
| +MoCo | $\mathbf{95.72}_{\pm0.05}$ | $\mathbf{95.60}_{\pm0.04}$ | $\mathbf{95.46}_{\pm0.03}$ | $\mathbf{95.23}_{\pm0.09}$ | $\mathbf{95.04}_{\pm0.20}$ | $\mathbf{73.90}_{\pm0.68}$ | $\mathbf{73.16}_{\pm0.57}$ | $\mathbf{72.69}_{\pm0.35}$ | $70.11_{\pm0.64}$ | $64.51_{\pm1.09}$ |
| +BYOL | $95.39_{\pm0.02}$ | $95.30_{\pm0.03}$ | $95.20_{\pm0.07}$ | $94.99_{\pm0.14}$ | $85.19_{\pm0.32}$ | $70.58_{\pm0.12}$ | $69.80_{\pm0.09}$ | $68.60_{\pm0.18}$ | $66.95_{\pm0.35}$ | $63.32_{\pm0.61}$ |

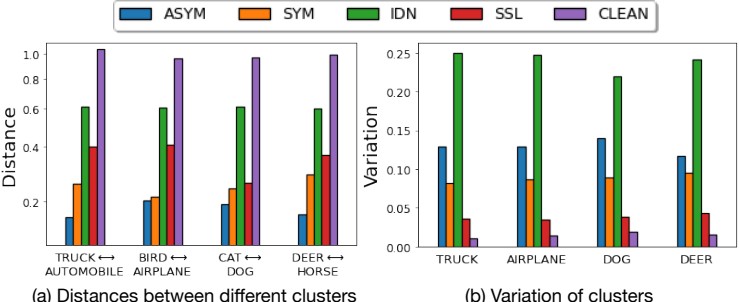

(a) Distances between different clusters  (b) Variation of clusters

Figure 3: Illustration of cluster structures for CIFAR-10 dataset. Representations are learned in different label noise settings. ASYM (blue): 40% asymmetric noise; SYM (orange): 60% symmetric noise; IDN (green): 60% instance-dependent noise. And we visualize distances between two clusters in (a) and the variance of each cluster in (b). The cluster structure (purple) serves as a baseline for representations that are trained by supervised learning without label noise.

**Cluster Structure.** We evaluate our cluster structure of SSL representations learned by MoCov2 on CIFAR-10. Learning SSL representations do not leverage the label information, so the representations are invariant to label noise, whereas representations learned by supervised learning (SL) are sensitive to label noise. We compare the cluster structure of SSL to that of SL in different label noise settings in Figure 3. We find that SSL representations (red) have a better cluster structure than SL representations obtained with different label noise. We note that although the distances between clusters of SL representations (green) learned with IDN are slightly larger than that of SSL representations, the variance of each cluster is 10 times larger. We also highlight the baseline cluster structure with purple, which is trained by the SL method without label noise.

**Fine-tuned Performance.** Following Chen et al. (2020b), we fine-tune the MoCov2 representation network on CIFAR-10 by GCE algorithm. For noise-free classification tasks, fine-tuning usually outperforms linear evaluation on various classification datasets (Chen et al., 2020b; Grill et al., 2020). However, Figure 4

Table 5: Test accuracy on CIFAR-10 semantic label noise over different noise levels.

| Dataset | TYPE-1 | | | | | TYPE-2 | | | | |
|---|---|---|---|---|---|---|---|---|---|---|
| Noise Level | 20% | 40% | 60% | 80% | 90% | 10% | 20% | 30% | 40% | 45% |
| GCE | $88.61_{\pm0.11}$ | $79.01_{\pm0.36}$ | $68.77_{\pm0.43}$ | $53.24_{\pm0.82}$ | $33.66_{\pm0.74}$ | $85.31_{\pm0.08}$ | $76.08_{\pm0.22}$ | $70.44_{\pm0.34}$ | $64.71_{\pm0.63}$ | $57.8_{\pm0.47}$ |
| +MoCo | $\mathbf{95.39}_{\pm0.04}$ | $\mathbf{94.80}_{\pm0.10}$ | $\mathbf{89.94}_{\pm0.16}$ | $\mathbf{83.13}_{\pm0.95}$ | $\mathbf{71.84}_{\pm0.59}$ | $\mathbf{90.92}_{\pm0.05}$ | $83.10_{\pm0.12}$ | $\mathbf{76.41}_{\pm0.17}$ | $69.60_{\pm0.29}$ | $66.68_{\pm0.55}$ |
| +BYOL | $94.86_{\pm0.04}$ | $93.69_{\pm0.31}$ | $89.19_{\pm0.36}$ | $\mathbf{83.53}_{\pm1.26}$ | $\mathbf{75.77}_{\pm1.44}$ | $89.64_{\pm0.27}$ | $\mathbf{83.34}_{\pm0.29}$ | $75.65_{\pm0.37}$ | $69.32_{\pm0.15}$ | $66.86_{\pm0.34}$ |
| CT | $91.06_{\pm0.33}$ | $72.78_{\pm0.35}$ | $44.30_{\pm0.47}$ | $25.37_{\pm0.18}$ | $17.30_{\pm0.23}$ | $85.52_{\pm0.12}$ | $76.11_{\pm0.12}$ | $65.36_{\pm0.20}$ | $55.54_{\pm0.81}$ | $48.64_{\pm0.38}$ |
| +MoCo | $\mathbf{95.02}_{\pm0.43}$ | $89.84_{\pm0.67}$ | $84.70_{\pm0.96}$ | $73.34_{\pm1.84}$ | $59.22_{\pm3.11}$ | $89.00_{\pm0.45}$ | $83.23_{\pm0.77}$ | $\mathbf{76.55}_{\pm0.92}$ | $\mathbf{72.41}_{\pm1.53}$ | $\mathbf{66.66}_{\pm1.98}$ |
| +BYOL | $94.58_{\pm0.07}$ | $\mathbf{91.56}_{\pm0.42}$ | $\mathbf{86.19}_{\pm0.70}$ | $73.27_{\pm0.61}$ | $\mathbf{64.75}_{\pm1.95}$ | $\mathbf{91.06}_{\pm0.47}$ | $83.10_{\pm0.20}$ | $76.50_{\pm0.18}$ | $68.57_{\pm1.23}$ | $63.82_{\pm1.58}$ |
| ELR | $94.19_{\pm0.05}$ | $91.75_{\pm0.51}$ | $82.72_{\pm0.43}$ | $70.86_{\pm0.46}$ | $39.05_{\pm0.72}$ | $86.69_{\pm0.02}$ | $79.06_{\pm0.08}$ | $71.02_{\pm0.11}$ | $62.09_{\pm0.27}$ | $58.02_{\pm0.26}$ |
| +MoCo | $\mathbf{95.76}_{\pm0.01}$ | $94.96_{\pm0.26}$ | $84.30_{\pm0.59}$ | $\mathbf{79.99}_{\pm0.45}$ | $63.63_{\pm1.09}$ | $88.27_{\pm0.49}$ | $84.53_{\pm0.27}$ | $\mathbf{78.45}_{\pm0.61}$ | $\mathbf{77.57}_{\pm0.35}$ | $69.27_{\pm1.70}$ |
| +BYOL | $95.66_{\pm0.02}$ | $\mathbf{95.35}_{\pm0.07}$ | $\mathbf{88.91}_{\pm0.31}$ | $77.83_{\pm0.42}$ | $\mathbf{76.98}_{\pm0.60}$ | $\mathbf{89.41}_{\pm0.87}$ | $\mathbf{85.83}_{\pm0.70}$ | $78.30_{\pm0.55}$ | $77.10_{\pm0.62}$ | $\mathbf{71.64}_{\pm1.50}$ |

Table 6: Test accuracy on ANIMAL-10N and Clothing1M

| Dataset | Animal-10N | | | | Clothing-1M | | | |
|---|---|---|---|---|---|---|---|---|
| Method | GCE | CT | ELR | TCL | GCE | CT | ELR | TCL |
| Origin | 84.58 | 86.93 | 86.52 | 87.90 | 71.34 | 71.68 | 71.89 | 73.81 |
| +MOCO | **87.35** | **87.66** | **88.51** | **89.51** | **72.61** | **72.41** | **72.71** | **74.31** |
| +BYOL | **88.42** | **88.36** | **88.68** | **89.43** | **72.90** | **72.63** | **72.98** | **74.38** |

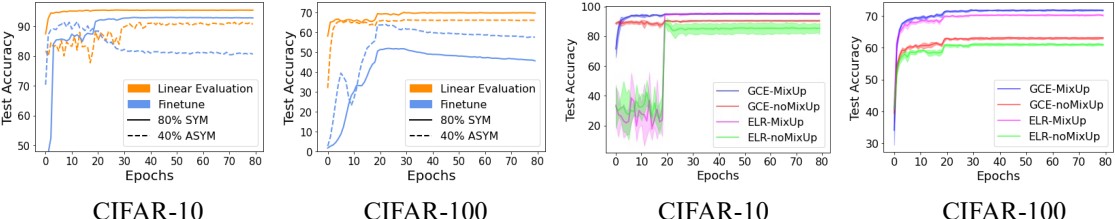

CIFAR-10    CIFAR-100    CIFAR-10    CIFAR-100

Figure 4: Left Two: comparison of linear eval and fine-tune on GCE. Right Two: comparison of SSL methods with and without MixUp component enabled on 80% symmetric label noise.

illustrates that linear evaluation (orange) performs better than fine-tuning (blue) in the presence of label noise. When label noise exits and the representations are not frozen, fine-tuning degrades the performance of the neural network. We hypothesize that fine-tuning the learned SSL representations destroys the cluster structure regarding true labels, leading to poor performance.

**The effects of MixUp augmentation.** We study the importance of mixup data augmentations. Our analysis indicates the importance of keeping larger $\delta$ in Lemma 7.3 and Proposition 7.4 by data augmentation. Without MixUp, the Definition 7.1 holds with smaller $\delta$. With MixUp enabled, the bound in Lemma 7.3 is tighter and the negative effects in Proposition 7.4 are mitigated. Results in Figure 4 indicate that applying MixUp augmentation significantly improves the performance on noisy datasets.

# 9 Conclusion

We provide a simple but effective method to address label noise. We first construct a motivating example to theoretically show that the classifier learned on SSL representations generalizes better than that from supervised learning. By further investigating the SSL representations under label noise, we find that: (1) The label noise is uniformly distributed over the data representations. (2) Representations learned by SSL exhibit good cluster properties, which encourages the linear classifier to be aligned with the optimal classifier. From the algorithmic perspective, we show that SSL representations can be applied as a strong complementary to various label noise methods by extensive experiments.

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
