## A   Experiment Details

### A.1   Noise Generation

For symmetric label noise, we randomly select a proportion of examples and then flip their labels to all possible labels with equal probabilities. Following Chen et al. (2021a), for asymmetric label noise in CIFAR-10, we randomly select a proportion of examples and flip their labels between TRUCK→AUTOMOBILE, BIRD→AIRPLANE, DEER→HORSE, and CAT↔DOG. For asymmetric label noise in CIFAR-100, we also randomly select a proportion of examples and flip their labels into the next class circularly. For instance-dependent label noise, we follow the intuition that mislabeled images share visually similar patterns (Xiao et al., 2015). To this end, we randomly choose an anchor image for each class, then we choose some similar images to the anchor image and flip their labels like symmetric label noise, where the similarity is measured by the L2 norm. To demonstrate this instance-dependent label noise is reasonable, we visualize a part of images from CIFAR-10 with similar visual patterns in Figure 5.

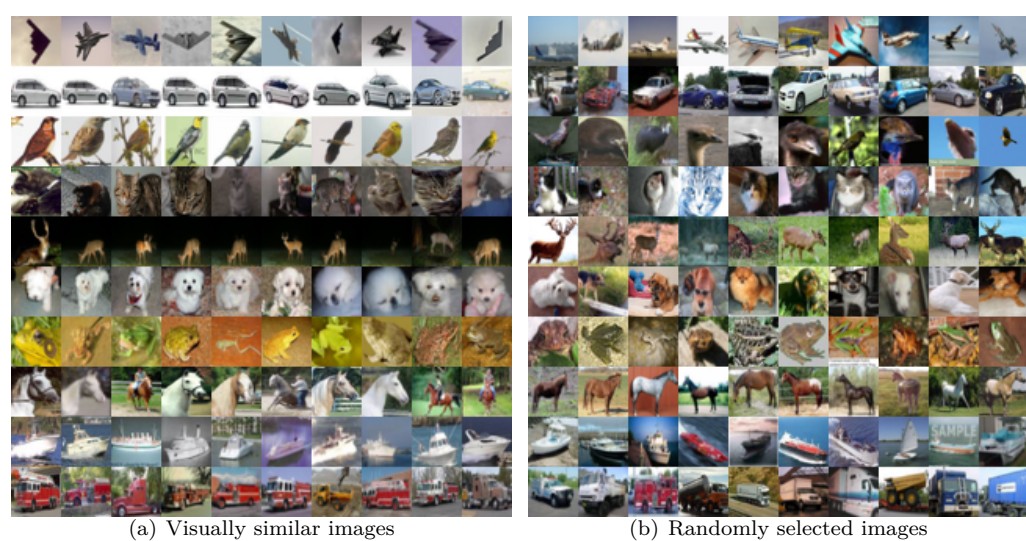

(a) Visually similar images                  (b) Randomly selected images

Figure 5: We artificially corrupted images with similar visual patterns. More specifically, we first randomly choose an anchor point for each class, shown in the first column of (a), and then we corrupted images that are similar to these anchor images. In contrast, (b) shows the randomly selected images for each class, which do not share similar patterns.

## B   Proofs for Theorem 3.1 and Proposition 4.1

**Lemma B.1.** *We consider the probability that for any interval $I_\Delta \subset [-m, b-m]$ with the length $\Delta = b(1 - e^{-d^{-5}})$, there exists at least one of those $\{\xi_i\}_{i=1}^n$ are within the interval $I_\Delta$. When $n = \text{Poly}(d)$, the statement holds with probability at least $1 - e^{-nd^5}$.*

*Proof.* Calculating event that there exist at least one of those $\{\xi_i\}_{i=1}^n$ are within the interval $I_\Delta$ is equivalent to calculate the event that all $\{\xi_i\}_{i=1}^n$ are not within the interval $I_\Delta$. Specifically

$$\Pr[\text{At least one}] = 1 - \Pr[\text{None}].$$

$$\Pr[\text{None}] = (\Pr[\xi_i \notin I_\Delta])^n \leq [\frac{b - \Delta}{b}]^n = [e^{-d^{-5}}]^n = e^{-nd^{-5}}.$$

Since all $\{\xi_i\}_{i=1}^n$ are sampled independently, we let $n = d^{10}$ without the loss of generality, then $\Pr[\text{At least one}] \geq 1 - e^{-d^5}$. We use this probability for the following proof.  □

**Proof for Theorem 3.1**

*Proof.* With the similar proof for $\{\zeta_i\}_{i=1}^n$, the conclusion of this Lemma B.1 also holds for $\{\zeta_i\}_{i=1}^n$ in a different interval $I_\Delta \subset [a, b]$. We can adopt the probability stated above for our proof. As pointed out in Soudry et al. (2018), the gradient descent with logistic loss over a linearly separable data induces a maximum L2 margin solution. It indicates that the induced classifier separates the data with respect to noisy labels and also satisfies the maximum L2 margin. Specifically, $\tilde{\omega} = \frac{\omega^\star}{\|\omega^\star\|}$, where $\omega^\star$ is given by:

$$\omega^\star = \underset{\omega \in \mathbb{R}^d}{\operatorname{argmin}} \|\omega\|^2 \text{ s.t. } \forall i : \omega^\top x_i \tilde{y}_i \geq 1. \tag{7}$$

The normalized optimal solution $\tilde{\omega} \in \{a_1 e_1 + a_2 e_2 : a_1 \in \mathbb{R}, a_2 \in \mathbb{R}\}$ since $a_3 e_3 + \cdots + a_d e_d$ is orthogonal to data point $x_i$ for any $a_j \in \mathbb{R}$, $j \in \{3, 4, \cdots, d\}$ and any $i \in [n]$. Let $\tilde{\omega} = \tilde{a}_1 e_1 + \tilde{a}_2 e_2$.

By Lemma B.1 with probability at least $1 - 2e^{-d^5}$ (this lower probability is adopted for simplicity), there exists a data point $(x_1 = e_1 \zeta_1 + e_2 \xi_2, \tilde{y}_1)$ where $|\zeta_1| \in [b - \Delta, b], |\xi_1| \in [0, \Delta]$. We analyze the case where $\zeta_1 > 0, \xi_1 > 0$. The analysis for other cases $\zeta_1 < 0, \xi_1 > 0$, $\zeta_1 > 0, \xi_1 < 0$, and $\zeta_1 < 0, \xi_1 < 0$ are similar. If $\zeta_1 > 0, \xi_1 > 0$, then $\tilde{y}_1 = 1$.

Consider the worst data point with $\zeta_1 = b - \Delta$ and $\xi_1 = \Delta$ that decides the lowest prediction accuracy of the classifier. To classify this point into the cluster $\tilde{y} = 1$, we need at least $\tilde{a}_1 > \frac{-\Delta}{\sqrt{\Delta^2 + (\Delta - b)^2}}$ and $\tilde{a}_2 > \frac{b - \Delta}{\sqrt{\Delta^2 + (\Delta - b)^2}}$. If there are other data points in this region, then the lower bound for $\tilde{a}_1$ is higher than $\frac{-\Delta}{\sqrt{\Delta^2 + (\Delta - b)^2}}$ and the lower bound for $\tilde{a}_2$ is higher than $\frac{b - \Delta}{\sqrt{\Delta^2 + (\Delta - b)^2}}$. Similarly, for other three cases, we have that $|\tilde{a}_1| < \frac{\Delta}{\sqrt{\Delta^2 + (\Delta - b)^2}}$ and $\tilde{a}_2 > \frac{b - \Delta}{\sqrt{\Delta^2 + (\Delta - b)^2}}$. Therefore, area of region misclassified by the classifier $\tilde{\omega}$ is at most $14 + \frac{16\Delta}{b - \Delta}$ (the total area is 32). Since the joint distribution of $X \times Y$ is uniform over the support, then the probability that a data is misclassified (in the misclassified region) is at most $\frac{7}{16} + \frac{\Delta}{2(4 - \Delta)}$, which is upper bounded by $\frac{7}{16} + \frac{\Delta}{6}$. Thus, the probability $\left| \operatorname{Pr}_{(x,y)}[\operatorname{sign}(\tilde{\omega}^\top x) = y] - 0.5625 \right| \leq \frac{2}{3}(1 - e^{-d^{-5}}) \leq \frac{2}{3d^5}$.

For self-supervised learning, as it is shown in Liu et al. (2021), the minimal solution $W_{\text{SSL}}$ to Eq. (1) is equivalent to the minimal solution to minimize $\left\| M - W^\top W \right\|_F^2$, where $M = \frac{1}{n} \sum_{i=1}^n x_i x_i^\top$. In the meanwhile, by Eckart-Young-Mirsky theorem, the span of $W_{\text{SSL}}$ is the top 1 eigenvector of $M$. Let the top 1 eigenvector be $e = p e_1 + q e_1$ with $p^2 + q^2 = 1$. The corresponding eigenvalue is given by:

$$
\begin{aligned}
e^\top M e &= \frac{1}{n} \sum_{i=1}^n \left[ (p e_1^\top + q e_2^\top)(\zeta_i e_1 + \xi_i e_2)(\zeta_i e_1^\top + \xi_i e_2^\top)(p e_1 + q e_2) \right] \\
&= \frac{1}{n} \sum_{i=1}^n \left[ p^2 \zeta_i^2 + q^2 \xi_i^2 + 2pq \zeta_i \xi_i \right] \\
&= \frac{1}{n} \sum_{i=1}^n \left[ (1 - q^2) \zeta_i^2 + q^2 \xi_i^2 + 2pq \zeta_i \xi_i \right] && (p^2 + q^2 = 1) \\
&= \frac{1}{n} \sum_{i=1}^n \zeta_i^2 - \frac{q^2}{n} \sum_{i=1}^n (\zeta_i^2 - \xi_i^2) + \frac{2pq}{n} \sum_{i=1}^n \zeta_i \xi_i \\
&= \frac{1}{n} \sum_{i=1}^n \zeta_i^2 - \frac{q^2}{n} \sum_{i=1}^n (\zeta_i^2 - \xi_i^2) + 2pq\mathbb{E}[\zeta \xi] + o(n^{-1/3}) \\
&= \frac{1}{n} \sum_{i=1}^n \zeta_i^2 - q^2 \underbrace{\frac{1}{n} \sum_{i=1}^n (\zeta_i^2 - \xi_i^2)}_{\text{①}} + o(n^{-1/3}).
\end{aligned}
$$

When $n$ is large enough, ① converges to its expectation. Let $\bar{Z} = \frac{1}{n} \sum_{i=1}^{n} (\zeta_i^2 - \xi_i^2)$. By Hoeffding's inequality

$$\mathbb{P}\{|\bar{Z} - \mathbb{E}[\bar{Z}]| \geq 1\} \leq 2\exp\left(-\frac{2n}{16^2}\right),$$

where $\mathbb{E}[\bar{Z}] = \frac{63}{16} \approx 3.93$. With probability at least $1 - 2e^{-n/128}$, $\bar{Z} \in [\frac{47}{16}, \frac{79}{16}]$ and $\bar{Z} \gg o(n^{-1/3})$. Therefore, the eigenvalue corresponding to the eigenvector $e$ is the largest when $q = 0$ and the top 1 eigenvector is $e = e_1$.

Without loss of generality, we let $W_{\text{SSL}} = e_1^\top$. After applying self-supervised transformation on inputs , the transformed data $\{(W_{\text{SSL}} x_i, \tilde{y}_i)\}_{i=1}^{n} = \{(y_i \zeta_i, \tilde{y}_i)\}_{i=1}^{n}$. We aim to learn any classifier where $\theta > 0$ to correctly predict all labels given inputs $\{y_i \zeta_i\}_{i=1}^{n}$ since $\zeta_i > 0, \forall i \in [n]$. The negative gradient of the logistic loss $\mathcal{L}(\theta)$ over $\{(y_i \zeta_i, \tilde{y}_i)\}_{i=1}^{n}$ is given by:

$$-\nabla_\theta \mathcal{L}(\theta) = \frac{1}{n} \sum_{i}^{n} \frac{\exp\left(-\tilde{y}_i y_i \zeta_i \theta\right)}{1 + \exp\left(-\tilde{y}_i y_i \zeta_i \theta\right)} \tilde{y}_i y_i \zeta_i.$$

Note that $\tilde{y}$ is independent with $\zeta$. And $\tilde{y}_i y_i = 1$ with probability $9/16$ and $\tilde{y}_i y_i = -1$ with probability $7/16$. Then with high probability, $-\nabla_\theta \mathcal{L}(\theta) > 0$. Eventually, there is a unique optimum $\theta = \theta_0 - \sum_i \alpha_i \nabla_\theta \mathcal{L}(\theta_i) > 0$ that minimizes the loss over $\{(y_i \zeta_i, \tilde{y}_i)\}_{i=1}^{n}$. When $\theta > 0$, the classifier gives the best decision boundary where $\text{sign}(\tilde{\theta} y_i \zeta_i) = y_i$. Hence, we have $\text{Pr}_{(x,y)}[\text{sign}(\tilde{\theta}^\top W_{\text{SSL}} x) = y] \geq 1 - 2e^{-n/128}$. □

**Proof for Proposition 4.1**

*Proof.* The proof for Proposition 4.1 can be adapted from the proof for Theorem 1. The optimal $W_{\text{SSL}}$ can be represented by the combination of $e_1$ and $e_2$. So we let $W_{\text{SSL}} = p_n e_1 + q_n e_2$, where $p_n$ and $q_n$ are optimal solutions dependent on the sample size $n$. By the definition of convergence in probability, we need to show

$$\text{Pr}[\|p_n e_1 + q_n e_2 - e_1\| > \epsilon] \to 0 \text{ as } n \to \infty,$$

for every $\epsilon > 0$. When $p_n = 1$ as $n \to \infty$, the above condition holds since $p_n + q_n = 1$. From the proof of Theorem 1, we have

$$\text{Pr}[p_n = 1] \geq 1 - 2e^{-n/128}.$$

Therefore

$$\begin{aligned}
\text{Pr}[\|p_n e_1 + q_n e_2 - e_1\| > \epsilon] &= \text{Pr}[\|p_n e_1 + q_n e_2 - e_1\| > \epsilon | p_n = 1] \text{Pr}[p_n = 1] \\
&\quad + \text{Pr}[\|p_n e_1 + q_n e_2 - e_1\| > \epsilon | p_n \neq 1] \text{Pr}[p_n \neq 1] \\
&\leq \text{Pr}[\|p_n e_1 + q_n e_2 - e_1\| > \epsilon | p_n = 1] + \text{Pr}[p_n \neq 1] \\
&\leq 2e^{-n/128} \to 0 \text{ as } n \to \infty
\end{aligned}$$

□

## C Proofs for Theorem 6.1

**Gradient Descent Discussion**

We discuss the behaviour of the linear classifier parameterized by $\omega_T$ given $-\widetilde{\nabla \mathcal{L}}(\omega_0)^\top \tilde{\mu} > \beta$, where $T$ is the time to stop training, and $0 < \beta \leq 1$. We first introduce the following lemma, which is used to characterize the convexity of the logistic loss over the data.

**Lemma C.1.** *Logistic loss $\mathcal{L}(\omega)$ is $\frac{\sigma_{\max}+1}{8}$-smooth when $n$ is large enough.*

*Proof.* The derivative of $\mathcal{L}(\omega)$ is given by:

$$\nabla \mathcal{L}(\omega) = \frac{1}{n} \sum_{i} (1 - \frac{1}{1 + \exp(-\tilde{y}_i \omega^\top x_i)})(-\tilde{y}_i x_i).$$

Then, the hessian of $\mathcal{L}(\omega)$ is given by:

$$\nabla(\nabla\mathcal{L}(\omega)) = \frac{1}{n}\sum_{i=1}^{n} x_i\sigma(i)(1-\sigma(i))x_i^\top,$$

where $\sigma(i) = 1/\big(1 + \exp(-\tilde{y}_i\omega^\top x_i)\big)$, and $\sigma_{\max}$ is the largest eigenvalue of $\Sigma$. Since $x \sim \mathcal{N}(\mathbf{0}, \Sigma/2)$, and by the rates of convergence for law of large numbers, we have

$$\nabla^2\mathcal{L}(\omega) \geq \frac{1}{4}(\Sigma/2 + o(n^{-1/3})) \preceq \frac{\sigma_{\max}+1}{8}.$$

$\square$

Based on the properties of smoothness:

$$(\nabla\mathcal{L}(\omega_{t+1}) - \nabla\mathcal{L}(\omega_t))^\top(\omega_{t+1} - \omega_t) \leq L\left\|(\omega_{t+1} - \omega_t)\right\|^2,$$

where $L = \frac{\sigma_{\max}+1}{8}$. Given that the descent algorithm,

$$\omega_{t+1} = \omega_t - \alpha_t\nabla\mathcal{L}(\omega_t)$$

By choosing appropriate learning rate $\alpha_t$ (for example $\alpha_t = \frac{1}{2L}$), we then have

$$\nabla\mathcal{L}(\omega_{t+1})^\top\nabla\mathcal{L}(\omega_t) \geq \frac{1}{4L}\left\|\nabla\mathcal{L}(\omega_t))\right\|^2 > 0.$$

Or equivalently,

$$\widetilde{\nabla\mathcal{L}}(\omega_{t+1})^\top\widetilde{\nabla\mathcal{L}}(\omega_t) \geq \underbrace{\frac{\left\|\nabla\mathcal{L}(\omega_t))\right\|}{4L\left\|\nabla\mathcal{L}(\omega_{t+1}))\right\|}}_{\gamma_t} > 0.$$

For two unit vectors, $\widetilde{\nabla\mathcal{L}}(\omega_0)$ and $\tilde{\mu}$, the higher cosine similarity means the lower L2 distance:

$$\left\|-\widetilde{\nabla\mathcal{L}}(\omega_0) - \tilde{\mu}\right\|^2 = 2 + 2\widetilde{\nabla\mathcal{L}}(\omega_0)^\top\tilde{\mu} \leq 2(1-\beta).$$

Similarly,

$$\left\|\widetilde{\nabla\mathcal{L}}(\omega_{t+1}) - \widetilde{\nabla\mathcal{L}}(\omega_t)\right\|^2 = 2 - 2\widetilde{\nabla\mathcal{L}}(\omega_{t+1})^\top\widetilde{\nabla\mathcal{L}}(\omega_t) \leq 2(1-\gamma_t).$$

Therefore,

$$\begin{aligned}
\left\|-\widetilde{\nabla\mathcal{L}}(\omega_1) - \tilde{\mu}\right\|^2 &= \left\|-\widetilde{\nabla\mathcal{L}}(\omega_1) + \widetilde{\nabla\mathcal{L}}(\omega_0) - \widetilde{\nabla\mathcal{L}}(\omega_0) - \tilde{\mu}\right\|^2 \\
&\leq 2\left\|\widetilde{\nabla\mathcal{L}}(\omega_1) - \widetilde{\nabla\mathcal{L}}(\omega_0)\right\|^2 + 2\left\|-\widetilde{\nabla\mathcal{L}}(\omega_0) - \tilde{\mu}\right\|^2 \\
&\leq 2(1-\gamma_0) + 2(1-\beta),
\end{aligned}$$

Equivalently,

$$-\widetilde{\nabla\mathcal{L}}(\omega_1)^\top\tilde{\mu} \geq 2(\beta + \gamma_0 - 1).$$

This can be easily generalized to the equation to time $t > 1$.

This can intuitively explain how larger $-\widetilde{\nabla\mathcal{L}}(\omega_0)^\top\tilde{\mu} > 0$ affects the performance of the classifier $\omega_T$. The conclusion with more rigorous justification can be found in Theorem 1 in Liu et al. (2020).

From another perspective, to intuitively understand why $-\widetilde{\nabla \mathcal{L}}(\omega_0)^\top \tilde{\mu} > \beta$ guarantees the behaviour of $\omega_T$ for $0 < \beta \leq 1$, we first decompose the gradient into two parts:

$$\nabla \mathcal{L}(\omega) = \frac{1}{n} \sum_i (1 - \frac{1}{1 + \exp(-\tilde{y}_i \omega^\top x_i)})(-\tilde{y}_i x_i)$$

$$= \frac{1}{n} \Big[ \underbrace{\sum_{i \in I_c} (1 - \frac{1}{1 + \exp(-\omega^\top x_i)})(-x_i)}_{\text{clean coefficients}} + \underbrace{\sum_{i \in I_n} (1 - \frac{1}{1 + \exp(\omega^\top x_i)})(x_i)}_{\text{mislabeled coefficients}} \Big],$$

where the density of $x$ is $\mathcal{N}(\mu, \Sigma)$, and $I_c$ is the index set of clean examples and $I_n$ is the index set of mislabeled examples. The first part is computed by weighted instances with clean labels, and we term the weights clean coefficients. Similarly, we term the weights for the second part mislabeled coefficients.

At the beginning ($t = 0$), clean samples dominate the gradient so we get the classifier closer to the optimal as $-\widetilde{\nabla \mathcal{L}}(\omega_0)^\top \tilde{\mu} > \beta$. Based on Proposition 5 in Liu et al. (2020), the clean coefficients decrease and the mislabeled coefficients increase as the training progresses. Eventually, they will achieve the balance, which leads to small $\nabla \mathcal{L}(\omega_t)$. Given that both the magnitude of $\nabla \mathcal{L}(\omega_t)$ and the learning rate $\alpha_t$ are small at time $t$, according to the gradient descent algorithm,

$$\omega_{t+1} = \omega_t - \alpha_t \nabla \mathcal{L}(\omega_t)$$

the learning will stop. Before that time, the learning is still dominated by clean examples and the performance of the classifier improves until convergence.

**Proof for Theorem 6.1**

*Proof.* By the rates of convergence for the law of large numbers

$$\nabla \mathcal{L}(\omega) = \frac{1}{n} \sum_i (1 - \frac{1}{1 + \exp(-\tilde{y}_i \omega^\top x_i)})(-\tilde{y}_i x_i)$$

$$= \mathbb{E}[\nabla \mathcal{L}(\omega)] + o(n^{-1/3}).$$

In this case of the symmetric label noise, the label noise function $\beta = -1$ with probability $r$ and $\beta = 1$ with probability $1 - r$, where $r$ controls the noise level. We decompose the expected gradient into the following form

$$\mathbb{E}[\nabla \mathcal{L}(\omega_0)] = \mathbb{E}[\mathbb{E}[\nabla \mathcal{L}(\omega_0)]|Y, \beta]$$

$$= \frac{1-r}{2} \mathbb{E}[\nabla \mathcal{L}(\omega_0)|Y = 1, \beta = 1] + \frac{r}{2} \mathbb{E}[\nabla \mathcal{L}(\omega_0)|Y = 1, \beta = -1]$$

$$+ \frac{1-r}{2} \mathbb{E}[\nabla \mathcal{L}(\omega_0)|Y = -1, \beta = 1] + \frac{r}{2} \mathbb{E}[\nabla \mathcal{L}(\omega_0)|Y = -1, \beta = -1],$$

where the derivative of $\mathcal{L}(\omega)$ is given by:

$$\nabla \mathcal{L}(\omega) = \frac{1}{n} \sum_i (1 - \frac{1}{1 + \exp(-\tilde{y}_i \omega^\top x_i)})(-\tilde{y}_i x_i). \tag{8}$$

To simplify the mathematical derivation, we assume that $\omega_0$ is initialized at 0. Based on this, the expected gradient can be simplified:

$$\mathbb{E}[\nabla \mathcal{L}(\omega_0)] = \frac{r}{2} \mathbb{E}[X|Y = 1, \beta = -1] - \frac{1-r}{2} \mathbb{E}[X|Y = 1, \beta = 1]$$

$$= (r - \frac{1}{2}) \mathbb{E}[X].$$

And,

$$\mathbb{E}[\nabla\mathcal{L}(\omega_0)^\top \mu] = (r - \frac{1}{2})\left\| \mu \right\|^2 \tag{9}$$

Then we compute $\left\| \nabla\mathcal{L}(\omega_0) \right\|$. By Jensen's inequality,

$$\begin{aligned}
\left\| \nabla\mathcal{L}(\omega_0) \right\| =& \frac{1}{2}\left\| \frac{1}{n}\sum_i \delta_i x_i \right\| && \delta_i \text{ is either } +1 \text{ or } -1\\
\leq& \frac{1}{2n}\sum_i \left\| x_i \right\|\\
\leq& \frac{1}{2}\sqrt{\left\| \mu \right\|^2 + c\,\mathrm{Trace}(\Sigma)},
\end{aligned}$$

where the last inequality is by the concentration property of sub-gaussian random vector and $c > 0$ is a constant.

Since $-\nabla\mathcal{L}(\omega_0)^\top \mu > 0$ by Eq. (9) given a sufficient number of examples, the condition $-\frac{\nabla\mathcal{L}(\omega_0)^\top \mu}{\left\| \nabla\mathcal{L}(\omega_0) \right\|\left\| \mu \right\|}$ is then given by:

$$\begin{aligned}
-\frac{\nabla\mathcal{L}(\omega_0)^\top \mu}{\left\| \nabla\mathcal{L}(\omega_0) \right\|\left\| \mu \right\|} =& \frac{(\frac{1}{2} - r)\left\| \mu \right\|^2 + o(n^{1/3})}{\left\| \nabla\mathcal{L}(\omega_0) \right\|\left\| \mu \right\|}\\
\geq& \frac{(1 - 2r)\left\| \mu \right\|}{\sqrt{\left\| \mu \right\|^2 + c\,\mathrm{Trace}(\Sigma)}} + o(n^{1/3}).
\end{aligned}$$

When the label noise is asymmetric, the results are the same though the mathematical expression is slightly different. For asymmetric label noise, we denote the label noise function $\beta(y) = -1$ with probability $r$ when $y = -1$, $\beta(y) = 1$ with probability $1 - r$ when $y = -1$, $\beta(y) = -1$ with probability $2r$ when $y = 1$, and $\beta(y) = 1$ with probability $1 - 2r$ when $y = 1$. Following similar derivations for the symmetric label noise,

$$\begin{aligned}
\mathbb{E}[\nabla\mathcal{L}(\omega_0)] =& \mathbb{E}[\mathbb{E}[\nabla\mathcal{L}(\omega_0)]|Y, \beta]\\
=& \frac{1 - r}{2}\mathbb{E}[\nabla\mathcal{L}(\omega_0)|Y = -1, \beta = 1] + \frac{r}{2}\mathbb{E}[\nabla\mathcal{L}(\omega_0)|Y = -1, \beta = -1]\\
& + \frac{1 - 2r}{2}\mathbb{E}[\nabla\mathcal{L}(\omega_0)|Y = 1, \beta = 1] + \frac{2r}{2}\mathbb{E}[\nabla\mathcal{L}(\omega_0)|Y = 1, \beta = -1]\\
=& \frac{3r - 1}{2}\mathbb{E}[X|Y = 1].
\end{aligned}$$

And,

$$\begin{aligned}
\mathbb{E}[\nabla\mathcal{L}(\omega_0)^\top \mu] =& \frac{3r - 1}{2}\Big[ \int_{-\infty}^{+\infty}(\left\| \mu \right\|^2 + W)\,\mathrm{d}\mathbb{P}_W \Big]\\
=& \frac{3r - 1}{2}\left\| \mu \right\|^2.
\end{aligned}$$

Therefore,

$$\begin{aligned}
-\frac{\nabla\mathcal{L}(\omega_0)^\top \mu}{\left\| \nabla\mathcal{L}(\omega_0) \right\|\left\| \mu \right\|} =& \frac{(\frac{1}{2} - \frac{3r}{2})\left\| \mu \right\|^2 + o(n^{1/3})}{\left\| \mathbb{E}[\nabla\mathcal{L}(\omega_0)] \right\|\left\| \mu \right\|}\\
\geq& \frac{(1 - 3r)\left\| \mu \right\|}{\sqrt{\left\| \mu \right\|^2 + c\,\mathrm{Trace}(\Sigma)}} + o(n^{1/3}).
\end{aligned}$$

$\square$

# D    Proofs for Lemma 7.2, Lemma 7.3, and Proposition 7.4

**Proof for Lemma 7.2**

*Proof.* The function $-\log \mathbb{E}[e^t]$ is concave since for any $t_1, t_2$, and $\alpha \in [0,1]$

$$-\log \mathbb{E}[e^{\alpha t_1 + (1-\alpha)t_2}] = -\log \mathbb{E}[(e^{t_1})^\alpha (e^{t_2})^{1-\alpha}]$$
$$= -\log \mathbb{E}[m_1^\alpha m_2^{1-\alpha}],$$

where $m_1 = e^{t_1}, m_2 = e^{t_2}$. By Holder's inequality, the above equality is further given by

$$-\log \mathbb{E}[m_1^\alpha m_2^{1-\alpha}] \geq -\log \left((\mathbb{E}[m_1])^\alpha (\mathbb{E}[m_2])^{1-\alpha}\right)$$
$$= -\alpha \log \mathbb{E}[e^{t_1}] - (1-\alpha)\log \mathbb{E}[e^{t_2}].$$

Therefore,

$$-\frac{1}{|J_i|} \sum_{k \in J_i} \log \mathbb{E}[e^{-\left\|f(u_k)-f(u_{g(k)})\right\|^2}] \leq -\log \mathbb{E}[e^{-\frac{1}{|J_i|}\sum_{k\in J_i}\left\|f(u_k)-f(u_{g(k)})\right\|^2}]$$

$$\leq -\log \mathbb{E}[e^{-\left\|\frac{1}{|J_i|}\sum_{k\in J_i}(f(u_k)-f(u_{g(k)}))\right\|^2}]$$

$$\leq -\mathbb{E}[\log e^{-\left\|\frac{1}{|J_i|}\sum_{k\in J_i}(f(u_k)-f(u_{g(k)}))\right\|^2}]$$

$$= \mathbb{E}\left[\left\|\frac{1}{|J_i|}\sum_{k\in J_i}(f(u_k)-f(u_{g(k)}))\right\|^2\right]$$

$$= \mathbb{E}[\|\mu_i - \mu_j\|^2].$$

Note that $-\|\cdot\|^2$ is concave and $-\log(\cdot)$ is convex. The proof is complete since

$$-\frac{1}{|J_i|} \sum_{k \in J_i} \log \mathbb{E}[e^{-\left\|f(u_k)-f(u_{g(k)})\right\|^2}] = -\frac{1}{|J_i|} \sum_{k \in J_i} \log \left(\mathcal{L}_{\text{Uniform}}(k, g(k))\right).$$

$\square$

**Proof for Lemma 7.3**

*Proof.* Let the region $R_{ij} = S_i \cap S_j$.

$$\mathbb{E}\left[\left\|f(u_i)-f(u_i^+)\right\|^2\right] = \mathbb{E}\left[\left\|f(u_i)-f(u_i^+)\right\|^2 | u_i^+ \in R_{ij}\right] \Pr[u_i^+ \in R_{ij}]$$
$$+ \mathbb{E}\left[\left\|f(u_i)-f(u_i^+)\right\|^2 | u_i^+ \notin R_{ij}\right] \Pr[u_i^+ \notin R_{ij}]$$
$$\geq \delta \mathbb{E}\left[\left\|f(u_i)-f(u_i^+)\right\|^2 | u_i^+ \in R_{ij}\right] \tag{10}$$

It shows that controlling the variance of a random variable controls the expected distance. For any different indices $i, j \in J_y$, with Eq.(10), we have for any $u_{ij} \in R_{ij}$

$$\mathbb{E}[\|f(u_i)-f(u_j)\|^2] \leq 2\mathbb{E}[\|f(u_i)-f(u_{ij})\|^2] + 2\mathbb{E}[\|f(u_j)-f(u_{ij})\|^2]$$
$$\leq \frac{2}{\delta}\mathbb{E}[\|f(u_i)-f(u_i^+)\|^2] + \frac{2}{\delta}\mathbb{E}[\|f(u_j)-f(u_j^+)\|^2], \tag{11}$$

where we omit the subscriptions for the expectation of simplicity when the context is clear.

The sample variance of the cluster $y$ is given by

$$\widehat{\Sigma}_y = \frac{1}{|J_y|} \sum_{i \in J_y} (f(u_i) - \frac{\sum_{j \in J_y} f(u_j)}{|J_y|})(f(u_i) - \frac{\sum_{j \in J_y} f(u_j)}{|J_y|})^\top$$

By the property $\mathrm{Trace}(AB) = \mathrm{Trace}(BA)$,

$$
\begin{aligned}
\mathrm{Trace}(\mathbb{E}[\widehat{\Sigma}_y]) &= \frac{1}{|J_y|^3} \mathbb{E} \sum_{i \in J_y} \left\| \sum_{j \in J_y} (f(u_i) - f(u_j)) \right\|^2 \\
&\leq \frac{1}{|J_y|^2} \mathbb{E} \sum_{i \in J_y} \sum_{j \in J_y} \| f(u_i) - f(u_j) \|^2 \\
&\leq \frac{2}{|J_y|\delta} \sum_{i \in J_y} \mathbb{E}[\| f(u_i) - f(u_i^+) \|^2] \qquad\qquad \text{by Eq. (11)} \\
&= \frac{2}{\delta|J_y|} \sum_{i \in J_y} \mathcal{L}_{\mathrm{Align}}(i)
\end{aligned}
$$

$\square$

**Proof for Proposition 7.4**

*Proof.* We assume $|J_y|$ for all $y \in \mathcal{Y}$ are the same as it is the most convenient and clear way to represent our results. We allow the different sizes for clusters and our results are unaffected. Note that $\log \mathbb{E}[e^t]$ is convex, following the spirits of proof from Lemma 5, we have

$$
\log\Big[\frac{1}{n(n-1)} \sum_{m \in \mathcal{Y}} \sum_{\substack{i,j \in J_m \\ i \neq j}} \mathcal{L}_{\mathrm{Uniform}}(i,j)\Big] \geq -\frac{1}{n(n-1)} \sum_{m \in \mathcal{Y}} \sum_{\substack{i,j \in J_m \\ i \neq j}} \mathbb{E}_{\substack{u_i \sim P(u|x_i) \\ u_j \sim P(u|x_j)}} \big[\, \| f(u_i) - f(u_j) \|^2 \,\big] \qquad (12)
$$

We replace terms in the right hand side of Eq. (12) associated with the the bound in Eq. (11), then we get

$$
\begin{aligned}
-\frac{1}{n(n-1)} \sum_{m \in \mathcal{Y}} \sum_{\substack{i,j \in J_m \\ i \neq j}} \mathbb{E}_{\substack{u_i \sim P(u|x_i) \\ u_j \sim P(u|x_j)}} \big[\, \| f(u_i) - f(u_j) \|^2 \,\big] &\geq -\frac{2}{\delta n(n-1)} \sum_{m \in \mathcal{Y}} (|J_m| - 1) \sum_{i \in J_m} \mathbb{E}[\| f(u_i) - f(u_i^+) \|^2] \\
&= -\frac{2(n/|\mathcal{Y}| - 1)}{\delta n(n-1)} \sum_{m \in \mathcal{Y}} \sum_{i \in J_m} \mathbb{E}[\| f(u_i) - f(u_i^+) \|^2] \\
&= -\frac{2(n/|\mathcal{Y}| - 1)}{\delta n(n-1)} \sum_{m \in \mathcal{Y}} \sum_{i \in J_m} \mathcal{L}_{\mathrm{Align}}(i)
\end{aligned}
$$

$\square$

# E   Hyperparameter analysis on augmentation strength

We conduct the sensitive analysis on different data augmentation strengths $\lambda = 1$ in Table 7 and 8. The results show that our method is stable under different augmentation strengths and shows consistent improvement over supervised learning.

Table 7: CIFAR-10 and CIFAR-100 datasets with IDN label noise over different augmentation strengths.

| Dataset | CIFAR-10 | | | | | CIFAR-100 | | | | |
|---|---|---|---|---|---|---|---|---|---|---|
| Noise | 20% | 40% | 60% | 80% | 90% | 20% | 40% | 60% | 80% | 90% |
| SSL+CE | 91.86 | 90.79 | 89.65 | 87.80 | 80.01 | 65.78 | 63.19 | 61.47 | 58.84 | 54.86 |
| $\lambda = 0.8$ MoCo+ELR | 94.61 | 94.43 | 94.31 | 95.51 | 89.15 | 70.41 | 69.73 | 68.23 | 63.21 | 57.42 |
| $\lambda = 0.9$ MoCo+ELR | 95.16 | 94.97 | 96.42 | 94.86 | 90.67 | 71.64 | 70.91 | 69.11 | 64.34 | 58.28 |
| $\lambda = 1.0$ MoCo+ELR | 95.77 | 95.70 | 95.65 | 95.58 | 91.35 | 72.74 | 71.56 | 69.69 | 65.94 | 59.80 |
| $\lambda = 1.1$ MoCo+ELR | 95.67 | 95.72 | 95.45 | 94.68 | 92.01 | 71.92 | 72.46 | 68.65 | 67.34 | 61.23 |
| $\lambda = 1.2$ MoCo+ELR | 95.21 | 94.98 | 96.23 | 94.88 | 92.20 | 70.64 | 70.46 | 70.79 | 67.13 | 60.21 |

Table 8: CIFAR-10 and CIFAR-100 datasets with IDN label noise over different augmentation strengths.

| Dataset | CIFAR-10 | | | | | CIFAR-100 | | | | |
|---|---|---|---|---|---|---|---|---|---|---|
| Noise | 20% | 40% | 60% | 80% | 90% | 20% | 40% | 60% | 80% | 90% |
| SSL+CE | 91.86 | 90.79 | 89.65 | 87.80 | 80.01 | 65.78 | 63.19 | 61.47 | 58.84 | 54.86 |
| $\lambda = 0.8$ BYOL+ELR | 93.31 | 91.78 | 92.29 | 96.27 | 91.21 | 73.71 | 71.74 | 69.53 | 62.19 | 56.43 |
| $\lambda = 0.9$ BYOL+ELR | 94.81 | 94.81 | 96.12 | 94.21 | 93.81 | 71.01 | 69.32 | 67.41 | 62.32 | 56.65 |
| $\lambda = 1.0$ BYOL+ELR | 95.45 | 95.25 | 95.08 | 95.07 | 94.91 | 72.11 | 70.64 | 68.72 | 63.75 | 57.54 |
| $\lambda = 1.1$ BYOL+ELR | 94.91 | 97.15 | 96.18 | 94.27 | 95.10 | 70.81 | 71.74 | 67.98 | 65.48 | 58.44 |
| $\lambda = 1.2$ BYOL+ELR | 95.31 | 94.10 | 94.98 | 94.71 | 95.10 | 71.10 | 72.64 | 69.42 | 65.32 | 57.14 |