# OpenReview forum: "Uniform Noise Distribution and Compact Clusters: Unveiling the Success of Self-Supervised Learning in Label Noise"
_TMLR — Accepted by TMLR_

### Review · Reviewer_xBbb · 2025-02-03

**Summary Of Contributions:**

This paper investigated how self-supervised learning (SSL) could address challenges posed by label noise in machine learning, especially instance-dependent noise. Most theoretical results were for a very specific synthetic noisy label data distribution, or only for a specific SSL method (Wang & Isola, 2020) not related to noisy label learning. Some empirical evidence on image datasets showing the benefits of SSL was presented.

**Audience:**

Yes

**Claims And Evidence:**

No

**Requested Changes:**

- The numberings of theorems are not consistent: Theorem 3.1 and Proposition 4.1 in the main body; Theorem 1 and Proposition 2 in the appendix. In appendix B, Lemma 8 and Lemma B.1 are just next to each other. You can use $\LaTeX$ packages `cleveref`, `thm-restate`, or `thmtools`.
- The theorems are not self-contained. The whole paragraphs before Theorem 3.1 are its assumptions. The same applies to Proposition 4.1 and Theorem 6.1.
- Section 5 is oddly short, and its claim is unclear.
- The notation should be rechecked. $\mathrm{Pr}[]$, $P()$, $P[]$, and $p()$ seem to be used interchangeably.
- Definition 7.1: Why it's asymmetric? Why $P[u | x_i]$ and not $P[u | x_j]$?

**Strengths And Weaknesses:**

## Strengths

- This paper aimed to theoretically analyze how self-supervised learning can address the label noise issue in machine learning, which is an important research question.
- This paper constructed a motivating example to demonstrate the benefits of self-supervised learning.
- This paper presented empirical evidence showing representations learned by self-supervised learning help improve robustness against noisy labels, especially instance-dependent noise.

## Weaknesses

- The definitions and theorems lack clarity and consistency (see below), significantly weakening the theoretical contributions.
- The first two theoretical results (Theorem 3.1 and Proposition 4.1) are very specific to a given problem: both the data distribution and the learning methods are fixed. Usually, stronger assumptions can lead to stronger theorems but weaker generalization ability. In this case, these results cannot even be generalized to any other data. All the design choices ($[-1.75, 2.25]$, $43.75%$) and results ($\frac{9}{16}$, $128$) are oddly specific because they are only applicable in this synthetic setting. Such theoretical results have little to no practical implications for real-world problems.
- The author claimed that SSL works because of "Uniform Noise Distribution: The label noise uniformly spreads over the learned SSL representation". However, I cannot see the correspondence between theoretical results and this claim.
- I cannot see how the analyses in Section 7 (Lemma 7.2, Lemma 7.3, and Proposition 7.4) are specific to noisy label learning. Even if they are true, they are just properties of SSL methods.

---

> ### Author Response · Authors · 2025-03-21
> **Response to review comments**
>
> We appreciate the constructive comments and insights on our work, which has substantially improved the quality of this manuscript. We have revised our manuscript and would like to address your concerns below:
>
> **1.	W1& W2: Practical implications of the theoretical results for real-world problems.**
>
> Thanks for the comments on the practical implications of our theoretical results, but we would like to clarify a potential misunderstanding.
>
> 1) The results in Sections 3 and 4 are used to present the intuition behind our theory in a more intuitive manner. In fact, as Reviewer FCwC pointed out, this is precisely one of the strengths of our work: “Sections 3-4 utilize a toy problem for label noise to show an advantage in terms of accuracy bounds for supervised learning with SSL over supervised learning alone.” Similar approaches have also been adopted in other studies, such as Invariant Risk Minimization [1] (e.g., Example 1 on page3 Section2).
>
> 2) ii.	In light of your comments, we provide more general results and practical implications about our theorem. We revised Section 5 to further highlight that the insights from Theorem 3.1 and Proposition 4.1 are empirically validated on real-world datasets such as Clothing1M (Figure 1 and Table 6). Concretely, Figure 1 (e,f) shows that Clothing1M trained with SSL has uniformly spread label noise and has higher accuracy in Table 6. This validates our analysis qualitatively and quantitatively.
>
> **2.  The correspondence between theoretical results and the claim**
>
> Thanks for raising this concern. The correspondence between theoretical results and our claim is discussed in Section 4, and we clarify the two points regarding the correspondence here.
>
> 1) Proposition 4.1 indicates that the SSL feature representation will make label noise dependent on the non-discriminative feature $e-2$ uniformly distributed. This makes the classifier learn the discriminative features $e_1$, which avoids learning the spurious correlation and improves the generalization.
>
> 2) Estimating the noise matrix becomes easier when label noise is uniformly distributed. Estimating the noise matrix is a commonly used approach in label noise learning.
>
> Besides, we have added a remark to show that Proposition 4.1 will make the label noise uniformly distributed over the representation. We also reorganized several paragraphs (page 5) to clarify the above two points.
>
> **3. How the analyses in Section 7 are specific to noisy label learning**
>
> Thank you for your comments. To better explain how the theoretical results in Section 7 is specific to noisy label learning, we first outline the overall logical flow of the paper.
> 1) Section 3 discusses why uniformly distributed label noise is beneficial.
> 2) Section 4 discusses how SSL satisfies such uniformly distributed noise.
> 3) Section 5 shows the generalization results in real-world data.
> 4) Section 6 discussed why a compact cluster is beneficial for label noise.
> 5) Section 7 discusses how SSL satisfies such properties.
> Basically, the logic is that we first demonstrate a beneficial property for learning label noise and then explain how SSL satisfies it. As for Section 7, it aims to explain how SSL satisfies the property required by Theorem 6.1 (as stated in the first sentence). Concretely, Definition 7.1 provides a tool to investigate how the SSL objective influences the cluster properties. Lemma 7.2 shows how SSL influences the cluster distance. Lemma 7.3 shows how SSL influences the cluster variance. Proposition 7.4 indicates the trade-off between these terms in Eq. 6, which explains the function of data augmentation in SSL training.
>
> We revised Section 7 and discussed the logic and relation of theorems and propositions of how SSL helps label noise learning.
>
> Reference:
> 1. Invariant Risk Minimization.
>
> For requested Changes:
>
> 1. We have unified the numbering system in the main submission and the supplementary.
>
> 2. We revised Theorem 3.1, Proposition 4.1, and Theorem 6.1 to make them self-contained.
>
> 3. We have revised Section 5 to detail the relation between theory and empirical evidence.
>
> 4. We have unified the notation to be $Pr[]$ for consistency.
>
> 5. The i and j in Definition 7.1 refer to two different instances' indexes. We use them to imply two different augmentations with the same class label.

---

### Review · Reviewer_FCwC · 2025-02-28

**Summary Of Contributions:**

This paper analyzes how SSL can assist with handling labeled noise for supervised learning. A toy problem is used to show how SSL compensates for label noise, leading to improved supervised learning accuracy when using frozen SSL features. This theoretical contribution motivates a comparison between supervised and SSL+supervised features, finding that SSL+supervised clusters data with respect to true labels and properly spreads noisy examples across classes. The clustering properties of the alignment and uniformity losses are then investigated, characterizing the mean and covariance of clusters with later empirical validation. Empirical evaluations investigate a variety of label noise setups on CIFAR-10 and CIFAR-100, as well as accuracy on two additional datasets.

**Audience:**

Yes

**Broader Impact Concerns:**

I do not have any broader impact concerns.

**Claims And Evidence:**

Yes

**Requested Changes:**

C1: [MoCo v3](https://arxiv.org/pdf/2104.02057) should be discussed in the SSL related works section in addition to the original MoCo paper.

C2: Where to find the proof of Theorem 1 should be stated near the end of Section 3.

C3: Proposition 2’s third sentence should change “depended” to “dependent”.

C4: It’s unclear whether ANIMAL-10 and Clothing-1M have label noise, so please specify in the “Datasets” part of Section 8.

**Strengths And Weaknesses:**

S1: Section 7 provides novel insights into the loss function proposed by Wang & Isola, investigating the mean and covariance of learned clusters and how these relate to the balance between alignment and uniformity losses.

S2: Empirical evaluations are very thorough. 4 datasets are used, with CIFAR-10 and CIFAR-100 being evaluated across varying types and strengths of label noise. A variety of methods are used and almost all results include some measure of variability, although to my knowledge it is not specified what this measure is and how it was calculated.

S3: Sections 3-4 utilize a toy problem for label noise to show an advantage in terms of accuracy bounds for supervised learning with SSL over supervised learning alone. This is then generalized to real data via Section 5, which shows via Figure 1 that the desired clustering according to true labels does occur.

W1: Lemma 8 (also called Lemma B.1?) has flaws in both sentences. As written, the first sentence is incorrect because there exists intervals such that no $\xi$ are within them. The proof calculates the probability of this occurrence. Therefore the first sentence should be reworded to not be a statement of fact, but a statement to be evaluated. For example, “We consider the probability that the statement ‘...’ is true.”

The second sentence only holds in the case where $n = d^{10}$, so this should either be stated in the lemma or the lemma should be generalized for $n$ equal to any polynomial of $d$.

W2: Theorem 3.1 (also called Theorem 1?) has 2 issues. I believe the first sentence should refer to $zeta$ and specify the differing range for the intervals.

“By Lemma B.1 with probability at least $1 - 2 e^{d^{-5}}$” is technically true, but I do not see why the true probability from applying the product rule, $(1 - e^{d^{-5}})^2$, is not used instead.

W3: The first paragraph of Section 7 misunderstands the relationship between prior SSL papers. Chen et al. is a precursor to He et al., Wang & Isola, and Tsai et al. Wang & Isola’s loss, given in Eq. 5, has not been widely adopted, to my knowledge, and is not cited in any of the papers used to support that claim. It is fine to focus on this loss, but its relevance should not be overstated as SimCLR variants are far more common.

---

> ### Author Response · Authors · 2025-03-21
> **Response to the review comments**
>
> Thank you for your constructive feedback! We appreciate that the reviewer found that our work provides novel insights with thorough empirical evaluations. Please find our response below:
>
> **1. Lemma B.1**
>
> We truly appreciate your constructive feedback on Lemma 8. Your points are well taken and have been very helpful in improving our work. We have revised the lemma and theorem accordingly to ensure they are more well-reasoned and clearly stated. We generalize Lemma B.1 to any polynomial of $n$ and explain that the choice of $n=d^{10}$ does not lose the generality.
>
> Additionally, we have adjusted the numbering system to maintain consistency between the main text and the supplementary material.
>
>
> **2. Theorem 3.1**
>
> Thanks for pointing out the issues. We revised the Theorem to accurately refer to the relevant variables and specify the correct range for the intervals for $\zeta_i$. Regarding $ 1 - 2e^{d^{-5}}$, we adopted it to simplify our proof, but $(1 - e^{d^{-5}})^2$ can also be used. We have added an explanation for the simplified choice.
>
> **3. The relationship between prior SSL papers**
>
> Thanks for the detailed advice. We have revisited the writing and avoided the overstated relevance of these papers in Section 7.
>
>
> **For requested changes:**
>
> 1. We have included MoCo v3 in the related work discussion, highlighting its improvements over previous MoCo versions and its relevance to SSL in noisy label settings.
>
> 2. We have now explicitly stated where to find the proof of Theorem 3.1 near the end of Section 3 to improve clarity. Besides, We have also revised the theorem numbering system to make it consistent in the main submission and appendix.
>
> 3. The wording in Proposition 2 has been corrected ("depended" → "dependent").
>
> 4. We have added more background information, references, and explanations in the dataset section of Section 8 to clarify the presence of label noise in Clothing-1M and ANIMAL-10.

---

### Review · Reviewer_Njpr · 2025-03-08

**Summary Of Contributions:**

This paper presents a theoretical framework explaining why self-supervised learning (SSL) succeeds with label noise, especially instance-dependent noise. The authors propose that SSL mitigates noisy labels by creating uniform noise distribution and forming compact, well-separated clusters in representation space. Theoretical proofs show classifiers trained on SSL representations outperform supervised learning approaches, with provided generalization accuracy bounds. The authors enhance SSL with mixup augmentation and validate their findings through extensive experiments on CIFAR-10, CIFAR-100, ANIMAL-10N, and Clothing1M datasets, demonstrating robustness against various noise types.

**Audience:**

Yes

**Claims And Evidence:**

Yes

**Requested Changes:**

1. Add a discussion about the gap between the SSL objective (1) and common practices in SSL methods.
2. Add more results about the computational cost and hyperparameter sensitivity analysis.

**Strengths And Weaknesses:**

## Strength
1. The paper addresses an important problem in machine learning - understanding why SSL works well with noisy labels, which has both theoretical significance and practical applications.
2. The work combines theoretical rigor (e.g., Theorems 3.1 and 6.1) with algorithmic insights, offering a well-rounded contribution to the field.

## Weakness

1. Theoretical gaps: Equation (1) is a simplified SSL objective using linear transformations, while practical methods like SimCLR employ non-linear MLPs. This discrepancy between the theoretical model and practical implementations limits the direct applicability of the paper's insights to SSL.
2. Experimental analysis: It would be better if the authors can discuss the computational cost or hyperparameter sensitivity of SSL methods.

---

> ### Author Response · Authors · 2025-03-21
> **Response to the review comments**
>
> Thank you for your constructive feedback! We carefully followed all your suggestions to improve the clarity of our manuscript. Below is our detailed response to each point.
>
> **The gap between the SSL objective (1) and common practices in SSL methods**
>
> We sincerely appreciate your comment regarding the theoretical gap, but we would like to clarify a potential misunderstanding. Specifically, we note that the common practices in SSL methods are indeed using a linear classifier. For example, SimCLR (e.g., section 6: linear evaluation part) and MoCo (e.g., section 4.1 linear classification protocol) all train a linear classifier for the final evaluation of SSL representations. Thus, our theoretical analysis adopts the same setting and is consistent with the practical research works in SSL. The reason that we, along with other works such as SimCLR, MoCo, and BYOL in the literature, deliberately use a linear classifier instead of a nonlinear classifier is that our research focuses on evaluating the benefits directly from SSL representations. Nonlinear layers, on the other hand, introduce additional flexibility in modifying representations for downstream tasks, which deviates from the original research intent of SSL.
>
> In light of your comments, we have clarified this point in the last paragraph of Section 1 stating that we follow the same setting in SSL research such as SimCLR, MoCo, and BYOL, and adopt the linear classifier for theoretical analysis.
>
>
> **More results about the computational cost and hyperparameter sensitivity analysis**
>
> Thank you for your suggestion. We added the computational cost in the implementation part in Section 8.  We have provided additional hyperparameter sensitivity analysis, but due to page limitations, we have included it in the supplementary material section E.

---

> ### Comment · Action_Editor_kua8 · 2025-04-14
>
> Dear Reviewer Njpr,
>
> This submission has been with us too long. We are still waiting for your official recommendation. Can you post it ASAP?
>
> AE

---

### Decision · Action_Editor_kua8 · 2025-04-18

**Recommendation:** Accept with minor revision

**Comment:**

This submission made the first attempt to theoretically explain why self-supervised representation learning can help label-noise learning, focusing on the properties of the noise distribution in the learned representations. In the beginning, all reviewers have major concerns about the theoretical analysis; after the rebuttal, two reviewers commented that their concerns had been addressed. Specifically, two positive reviewers said the following in their official recommendations:
> This paper focus on an important problem of why SSL succeeds with label noise and combines theoretical rigor (e.g., Theorems 3.1 and 6.1) with algorithmic insights, offering a well-rounded contribution to the field. My concerns about the theoretical results and experiments are also addressed during rebuttal. Therefore, I recommend an acceptance for this work.

> The claims made in the submission supported by accurate, convincing and clear evidence and individuals in TMLR's audience interested in contrastive learning would enjoy the findings of this paper.

while the negative reviewer said
> It is unclear if the proposed theoretical results for a very specific problem are insightful in practice.

I feel that this paper fits TMLR well and we should accept it for publication. I don't know if there are still addressable concerns now (from Reviewer xBbb), so I think "accept with minor revision" may be better.

**Audience:**

yes

**Claims And Evidence:**

yes